# A small molecule inhibitor of PTP1B and PTPN2 enhances T cell anti-tumor immunity

Shuwei Liang[1,2,9], Eric Tran [3,9], Xin Du[1,2], Jiajun Dong[4], Harrison Sudholz [1,2], Hao Chen[5,6], Zihan Qu[7], Nicholas D. Huntington [1,2], Jeffrey J. Babon [5,6], Nadia J. Kershaw [5,6], Zhong-Yin Zhang [4,7], Jonathan B. Baell [3,8], Florian Wiede[1,2] ✉ & Tony Tiganis [1,2] ✉

The inhibition of protein tyrosine phosphatases 1B (PTP1B) and N2 (PTPN2) has emerged as an exciting approach for bolstering T cell anti-tumor immunity. ABBV-CLS-484 is a PTP1B/PTPN2 inhibitor in clinical trials for solid tumors. Here we have explored the therapeutic potential of a related small-molecule-inhibitor, Compound-182. We demonstrate that Compound-182 is a highly potent and selective active site competitive inhibitor of PTP1B and PTPN2 that enhances T cell recruitment and activation and represses the growth of tumors in mice, without promoting overt immune-related toxicities. The enhanced anti-tumor immunity in immunogenic tumors can be ascribed to the inhibition of PTP1B/PTPN2 in T cells, whereas in cold tumors, Compound-182 elicited direct effects on both tumor cells and T cells. Importantly, treatment with Compound-182 rendered otherwise resistant tumors sensitive to α-PD-1 therapy. Our findings establish the potential for small molecule inhibitors of PTP1B and PTPN2 to enhance anti-tumor immunity and combat cancer.

The advent of immunotherapy has revolutionized the management of hematologic and solid tumors[1]. In particular, antibodies that target immune checkpoints and alleviate inhibitory constraints imposed on T cells by tumors have changed the treatment landscape for cancer. Indeed, antibodies neutralizing the PD-1 (Programmed cell death protein 1) and CTLA-4 (cytotoxic T-lymphocyte-associated protein 4) checkpoints yield robust and durable responses for a variety of tumors, including melanomas, non-small cell lung carcinomas, and Hodgkin lymphoma[1,2]. However, for many tumors, response rates are low. Such tumors are typically characterized by the absence or paucity of immune infiltrates, especially T cells, and are referred to as non-immunogenic or "cold" tumors[1,3]. This paucity can be ascribed to various factors, including low tumor mutational burdens, poor T cell homing and infiltration and/or the downregulation of antigen presentation so that tumors are not detected by the immune system[4,5]. However, adaptive or acquired resistance to immunotherapy are also common, so that even tumors predominated by tumor-infiltrating lymphocytes (TILs) do not respond, whereas others initially respond, but then relapse[4,5]. One possible approach by which to overcome a number of these challenges is to target protein tyrosine phosphatases (PTPs) that not only antagonize T cell function, but can also directly modulate the way tumors engage the immune system[6–13].

In the last several years the classical tyrosine-specific phosphatases PTP1B (encoded by *PTPN1*) and PTPN2 (also known as TCPTP and encoded by *PTPN2*) have emerged as exciting immunotherapy targets for cancer[6–11,14]. PTP1B and PTPN2 are two of the most closely related members of the PTP superfamily, sharing a high degree of catalytic domain sequence and structural identity, but differing in their non-

[1]Monash Biomedicine Discovery Institute, Monash University, Clayton, Victoria 3800, Australia. [2]Department of Biochemistry and Molecular Biology, Monash University, Clayton, Victoria 3800, Australia. [3]Monash Institute of Pharmaceutical Sciences, Monash University, Parkville, Victoria 3052, Australia. [4]Department of Medicinal Chemistry and Molecular Pharmacology, Purdue University, West Lafayette, IN 47907, USA. [5]Walter and Eliza Hall Institute of Medical Research, Parkville, Victoria 3052, Australia. [6]Department of Medical Biology, The University of Melbourne, Melbourne, Victoria 3052, Australia. [7]Department of Chemistry, Purdue University, West Lafayette, IN 47907, USA. [8]Lyterian Therapeutics, South San Francisco, San Francisco, CA 94080, USA. [9]These authors contributed equally: Shuwei Liang, Eric Tran. ✉e-mail: Florian.Wiede@monash.edu; Tony.Tiganis@monash.edu

catalytic N- and C-terminal segments[15]. PTP1B is targeted to the cytoplasmic face of the endoplasmic reticulum (ER) by a hydrophobic C-terminus, but can access substrates at the plasma membrane and after receptor endocytosis[15–17]. On the other hand, PTPN2 can exist as two variants: a 48 kDa variant, which like PTP1B is targeted to ER by a hydrophobic C-terminus, and a 45 kDa variant that lacks the hydrophobic C-terminus and is targeted to the nucleus and shuttles between the nuclear and cytoplasmic environments[18]. The two PTPs can act together to regulate diverse biological processes, including for example the hypothalamic control of energy expenditure and glucose metabolism[19,20], as well as T cell biology and function[11,21–26] and they do so by dephosphorylating both distinct and overlapping substrates[18]. For example, both PTPs are key negative regulators of JAK (Janus-activated kinase)/STAT (signal transducer and activator of transcription) signaling, but PTP1B is selective for JAK-2 and Tyk2[27,28] and PTPN2 for JAK-1 and JAK-3 in the cytoplasm[29] and additionally STAT-1, STAT-3 and STAT-5 in the nucleus in a cell context-dependent manner[18,19,30–33].

In T cells, PTPN2 antagonizes T cell receptor (TCR) signaling, by dephosphorylating and inactivating the Src family kinases (SFKs) LCK and FYN and additionally tempers cytokine signaling, especially interleukin (IL)-2-induced STAT-5 and interferon (IFN)-induced STAT-1 signaling, to tune T cell responses and prevent inappropriate responses to self in the context of T cell homeostasis and antigen cross-presentation[7,21,22,25,34]. PTPN2's importance in T cell tolerance is substantiated by studies demonstrating that the conditional deletion of *Ptpn2* in T cells can result in overt autoimmunity in aged C57BL/6 mice[21], or accelerate the onset of type 1 diabetes and co-morbidities in autoimmune-prone NOD mice[34], whereas its global deletion[35] or inducible deletion in hematopoietic cells[25] can promote marked systemic inflammation and autoimmunity. These phenotypes are similar to those evident in mice in which the immune checkpoint receptors PD-1[36–38] or CTLA-4[39,40] have been deleted. In line with observations in mice, *PTPN2* loss of function single nucleotide polymorphisms (SNPs) in humans have also been associated with autoimmune and inflammatory diseases[41–44]. By contrast, although PTP1B affects T cell development and function by attenuating IL-2-/-5/-15-induced STAT-5 signaling, it does not regulate TCR signaling, and its deletion globally, or in T cells alone, does not promote systemic inflammation or autoimmunity[11]. Nevertheless, we and others have shown that the deletion of either PTP1B or PTPN2 in T cells can markedly enhance anti-tumor immunity[7,8,11,14]. PTPN2 deletion enhances T cell-mediated immune surveillance to prevent the formation of hematologic and solid malignancies that otherwise occur in aged mice heterozygous for the tumor suppressor *p53* and also enhances the anti-tumor activity of adoptively transferred T cells, including chimeric antigen receptor (CAR) T cells[14]. These effects are attributed to the promotion of LCK signaling, enhancing T cell/CAR T cell activation, as well as IL-2-induced STAT-5 signaling to promote CXCR3 expression and the homing of T cells to C-X-C Motif Chemokine Ligand 9 (CXCL9)- and CXCL10-expressing tumors and the acquisition of effector/cytotoxic functions[14]. Moreover, other studies have shown that the deletion of PTPN2 and the promotion of IFNAR1-STAT-1 signaling can also overcome T cell exhaustion[7]. On the other hand, we have shown that PTP1B levels are elevated in intratumoral CD8+ effector T cells isolated from human melanomas or syngeneic tumors in mice, and that the inhibition or deletion of PTP1B in T cells, or CAR T cells can markedly enhance anti-tumor immunity by promoting STAT-5 signaling[11]. These studies have identified PTP1B and PTPN2 as intracellular T cell checkpoints, with analogous functions to those mediated by the cell surface T cell inhibitory receptor PD-1.

Beyond their roles in T cells, PTP1B and PTPN2 also have cell autonomous roles in tumors, by directly influencing tumor growth, as well as the ability of tumors to interact with immune cells[6,10,45–48]. Several studies indicate that PTP1B might serve as a potential therapeutic target in solid tumors, especially HER2-positive breast cancer, where it can contribute to HER2 signaling and tumor growth[45–48]. On the other hand, the deletion of PTPN2 in tumors can enhance IFN-induced STAT-1 signaling to drive the expression of T cell chemoattractants, including CXCL9 and CXCL10, genes associated with antigen presentation, including major histocompatibility complex (MHC) class I (MHC-I), as well as genes encoding ligands for immune checkpoints, including programmed death ligand 1 (PD-L1)[6,10]. The deletion of PTPN2 in syngeneic tumors in mice, including xenografted B16F10A melanomas and MC38 colorectal adenocarcinomas, as well as orthotopic AT3 mammary tumors, can enhance T cell mediated anti-tumor immunity and the response to PD-1 checkpoint blockade[6,10]. These preclinical findings appear to be relevant to human tumors, since low PTPN2 protein in triple-negative breast cancer (TNBC) is accompanied with TILs/T cells and increased PD-L1 levels, whereas low *PTPN2* mRNA is associated with improved survival[10]. Importantly, using genetic approaches we have been able to show that the combined targeting of PTPN2 in tumor cells and T cells in mice can yield even greater anti-tumor immunity[10].

Preclinical studies have established the therapeutic potential of targeting PTP1B or PTPN2 for the promotion of anti-tumor immunity to combat cancer[6–9,11,14]. Indeed, such studies point towards the combined targeting of PTP1B or PTPN2 in tumor cells and T cells, and potentially other immune cells, eliciting synergistic outcomes[8–11]. At present two PTP1B/PTPN2 inhibitors, ABBV-CLS-484 (Fig. S1) and ABBV-CLS-579, are in phase I clinical trials (NCT04417465, NCT04777994) for patients with locally advanced or metastatic tumors and are being tested alone and in combination with α-PD-1. In this study, we sought to explore the efficacy and safety of small molecule inhibitors targeting PTP1B and PTPN2 in cancer.

## Results

### Synthesis and purity of Compound 182

To explore the therapeutic potential of targeting PTP1B and PTPN2 in cancer with a small molecule inhibitor, we took advantage of AbbVie's/Calico's Compound 182, which inhibits both PTP1B and PTPN2 and is related to the small molecule drug ABBV-CLS-484 (Supplementary Figs. 1 and 2) currently in clinical trials. The structure of AbbVie's/Calico's Compound 182 was originally described in patent application WO2019246513A1. To synthesize Compound 182 (**XI**), we adapted a synthetic route (Supplementary Fig. 2) from the previously disclosed patent WO2019246513A1. Commercially available 7-bromo-3-hydroxy-2-naphthoic acid **I** was reacted with excess benzyl bromide, followed by base-mediated hydrolysis to afford the free carboxylic acid **II**. A Curtius rearrangement then subsequent hydrolysis was used to afford amine **III**. Alkylation of the amine afforded compound **IV**, which was then fluorinated to give compound **V**. A preformed solution of sulfonyl chloride **VII** was reacted with amine **V**, then a deprotection resulted in the free sulfamide **VIII**. A base-mediated cyclisation resulted in the formation of the acylsulfamide **IX**. The aryl bromide was then converted to the corresponding phenol through a Pd-catalyzed reaction, which was subsequently alkylated with commercially available 4-bromo-2-methylbutan-2-ol in a two-step one-pot procedure to give compound **X**. Finally, removal of the benzyl-protecting group yielded **XI** (Compound 182), whose purity after reverse phase chromatography was 95% (Supplementary Fig. 3) and whose structure was confirmed by NMR spectroscopy (Supplementary Figs. 4 and 5).

### Compound 182 is a specific active site inhibitor of PTP1B and PTPN2

Next, we characterized Compound 182's mode of action using recombinant PTPs in vitro. Using 6,8-Difluoro-4-methylumbelliferyl phosphate (DiFMUP; 10 μM) as a substrate, Compound 182 inhibited the catalytic activities of PTP1B and PTPN2 with IC$_{50}$s of $0.63 \pm 0.01$ and $0.58 \pm 0.02$ nM respectively (Table 1). Kinetic analyses (Lineweaver-Burk) with varying inhibitor and DiFMUP substrate concentrations

**Table 1 | PTP selectivity of Compound 182**

| Enzyme | IC50, µM | Fold selectivity | Enzyme class |
|---|---|---|---|
| TC-PTP | 0.00058 ± 0.00002 | 1 | Non receptor PTPs |
| PTP1B | 0.00063 ± 0.00001 | 1.1 | |
| SHP-1 | 9.1 ± 0.4 | 15690 | |
| SHP-2 | 19.2 ± 1.7 | 33103 | |
| LYP | >>20 | >>34000 | |
| STEP | >>20 | >>34000 | |
| HePTP | >>20 | >>34000 | |
| PTP-PEST | >>20 | >>34000 | |
| FAP-1 | >>20 | >>34000 | |
| PTPα | >>20 | >>34000 | Receptor-type PTPs |
| PTPε | 0.8 ± 0.1 | 1379 | |
| CD45 | >>20 | >>34000 | |
| CDC14A | 6.3 ± 0.8 | 10862 | Dual-specificity PTPs |
| CDC14B | 5.2 ± 1.7 | 8966 | |
| MKP5 | >>20 | >>34000 | |
| VHZ | >>20 | >>34000 | |
| Laforin | >>20 | >>34000 | |
| LMPTP | >>20 | >>34000 | LMW PTP |

under steady state conditions established that Compound 182 is a classic competitive inhibitor of PTP1B and PTPN2, with $K_i$ values of 0.34 ± 0.05 and 0.26 ± 0.02 nM respectively (Fig. 1a). A jump-dilution assay (where Compound 182 was preincubated with PTP1B or PTPN2 and subsequently diluted into saturating substrate to wash off unbound compound, followed by the analysis of residual enzyme activity) established that Compound 182 inhibits PTP1B and PTPN2 in a reversible manner (Fig. 1b), whereas label-free differential scanning fluorescence (DSF) assays validated target engagement. Specifically, we found that the inflection temperatures ($T_i$) for PTP1B and PTPN2 were 56.8 ± 0.4 and 55.2 ± 0.3 °C, respectively (Fig. 1c). Compound 182 treatment increased the thermal stability of PTP1B and PTPN2 and resulted in positive thermal shifts with $\Delta T_i$s of 9.5 ± 0.5 and 17.4 ± 0.8 °C, respectively (Fig. 1c), hence validating binding between Compound 182 and the PTP1B or PTPN2 proteins.

To determine the selectivity of Compound 182 for PTP1B and PTPN2, we profiled its inhibitory action against 16 PTPs, including receptor-like classical PTPs (PTPα, PTPε, CD45), non-receptor-like classical PTPs (SHP-1, SHP-2, LYP/PTPN22, STEP, HePTP, PTP-PEST, FAP-1), dual specificity phosphatases (CDC14A, CDC14B, MKP5, VHZ, Laforin) and low molecular weight PTPs (LMPTP)[18]. Compound 182 exhibited >15,000-fold selectivity for PTP1B and PTPN2 over most of the PTPs screened with the exception of PTPε, CDC14B, and CDC14A for which it was >1300-, >8000- and >10,000-fold respectively more selective for PTP1B and PTPN2 (Table 1). Taken together, these results demonstrate that Compound 182 is a potent, reversible, and competitive inhibitor of PTP1B and PTPN2 with remarkable specificity.

Both Compound 182 and ABBV-CLS-484 contain a fluorinated phenolic acylsulfamide phosphotyrosine (pTyr) mimetic (Supplementary Fig. 1) likely to engage the PTP active site[49,50]. To gain insight into the molecular basis for Compound 182's selectivity for PTP1B and PTPN2 over other PTP family members, we solved the structure of PTP1B in complex with Compound 182 using X-ray crystallography. The diffraction data was processed to 1.55 Å (Table 2) and phases were obtained by molecular replacement using the structure of apo-PTP1B (PDB: 7MKZ). As might be anticipated, the pTyr-mimicking acylsulfamide was located in the active site as previously seen with core-related isothiazolidinone (IZD)-based pTyr mimetics[49,50] (Fig. 1d; Supplementary Figs. 6 and 7). Briefly, the phosphate position was occupied by the acylsulfamide ring with the carbonyl oxygen atom displacing a

conserved water molecule that is usually hydrogen bonded to Gln266[51]. This placed the edge of the acylsulfamide ring above the catalytic Cys215 in such a way that the hydrogen-bond network that normally exists between PTPs and the phosphate of their substrate pTyr was largely maintained. The central naphthalene moiety in the inhibitor overlaid with the phenyl-ring of the substrate pTyr and the 3-hydroxy-3-methylbutoxy tail then extended along the surface of the protein towards Gly259. The 3-hydroxyl group at the terminus of the compound was not quite within hydrogen bonding distance of Gln262. However, there was a network of ordered water molecules between them. The WPD loop of PTP1B was closed, as it is in previous structures with IZD-based inhibitors[50], placing the planar naphthalene core between Phe182 on the exterior of the protein and a hydrophobic surface made up of sidechains from Tyr46, Val49, Ile219, and Ala217. The naphthol group was hydrogen bonded to Asp181, the catalytic acid, whilst the fluorine atom was adjacent to Gln262, the residue thought to position the catalytic water for nucleophilic attack.

The PTP1B/Compound 182 structure reveals that the 3-hydroxy-3-methylbutoxy tail of Compound 182 is likely responsible for its specificity towards PTP1B and PTPN2. The terminal methyl group abuts Gly259 and anything larger than a Gly at this position would clash with Compound 182 (Fig. 1d). Likewise, the ether group at the beginning of the tail was 3.6 Å from Val49 and any longer sidechain (most PTPs contain Ile at this position) would also clash. Finally, the naphthalene core would clash with any reside larger than Ala at position 217. Since, amongst the 37 classical human PTPs, the combination of Ala217, Gly259, and Val49 is unique to PTP1B and PTPN2[52] we hypothesize that these residues are responsible for Compound 182's remarkable specificity.

**Compound 182 promotes TCR and cytokine signaling in T cells**
We have shown previously that the deletion of PTP1B or PTPN2 in T cells can enhance T cell activation and cytotoxicity in vitro by promoting TCR and/or cytokine signaling, whereas in vivo the deletion of PTPN2 or PTP1B in T cells can markedly enhance anti-tumor immunity[11,14]. Accordingly, we first sought to determine whether Compound 182 could enhance T cell signaling and function ex vivo (Fig. 2). Compound 182 enhanced the antigen-induced activation of CD8+ T cells, as assessed by measuring the cell surface levels of the activation markers CD25 and CD69 after challenging OT-I TCR transgenic CD8+ T cells with the cognate ovalbumin (OVA) peptide antigen SIINFEKL (N4) (Fig. 2a). This was accompanied by enhanced CD8+ T cell cytotoxic potential, as reflected by the increased expression of Granzyme B (GRZMB), IFNγ and TNF that mediate cytotoxic CD8+ T cell responses (Fig. 2b). The extent of T cell activation approximated that resulting from the deletion PTPN2 in T cells (Lck-Cre;Ptpn2^fl/fl). This enhanced antigen-induced T cell activation was in turn accompanied by marked increases in OT-I CD8+ T cell proliferation/expansion as reflected by the greater dilution of the cell-permeable dye CTV (Fig. 2c). Moreover, as might be expected from its ability to inhibit both PTP1B and PTPN2, the inhibitor additionally enhanced the antigen-induced activation and expansion of PTPN2-deficient (Lck-Cre;Ptpn2^fl/fl) CD8+ T cells (Fig. 2c). Finally, in line with previous studies that have shown that PTPN2 and PTP1B act to negatively regulate TCR-induced LCK signaling and/or cytokine signaling respectively[11,14], we found that Compound 182 enhanced TCR-induced (α-CD3/IgG cross-linking) SFK Y418 phosphorylation (p-SFK) (Fig. 2d) as well as basal and IL-2-induced STAT-5 Y694 phosphorylation (p-STAT-5) (Fig. 2e, f) in CD8+ T cells. Taken together these results demonstrate that Compound 182 can significantly enhance T cell responses ex vivo.

**Compound 182 represses tumor growth**
Having established that Compound 182 can enhance T cell activation ex vivo we next assessed its impact on anti-tumor immunity in vivo. To this end, we implanted immunogenic OVA-expressing AT3 (AT3-OVA)

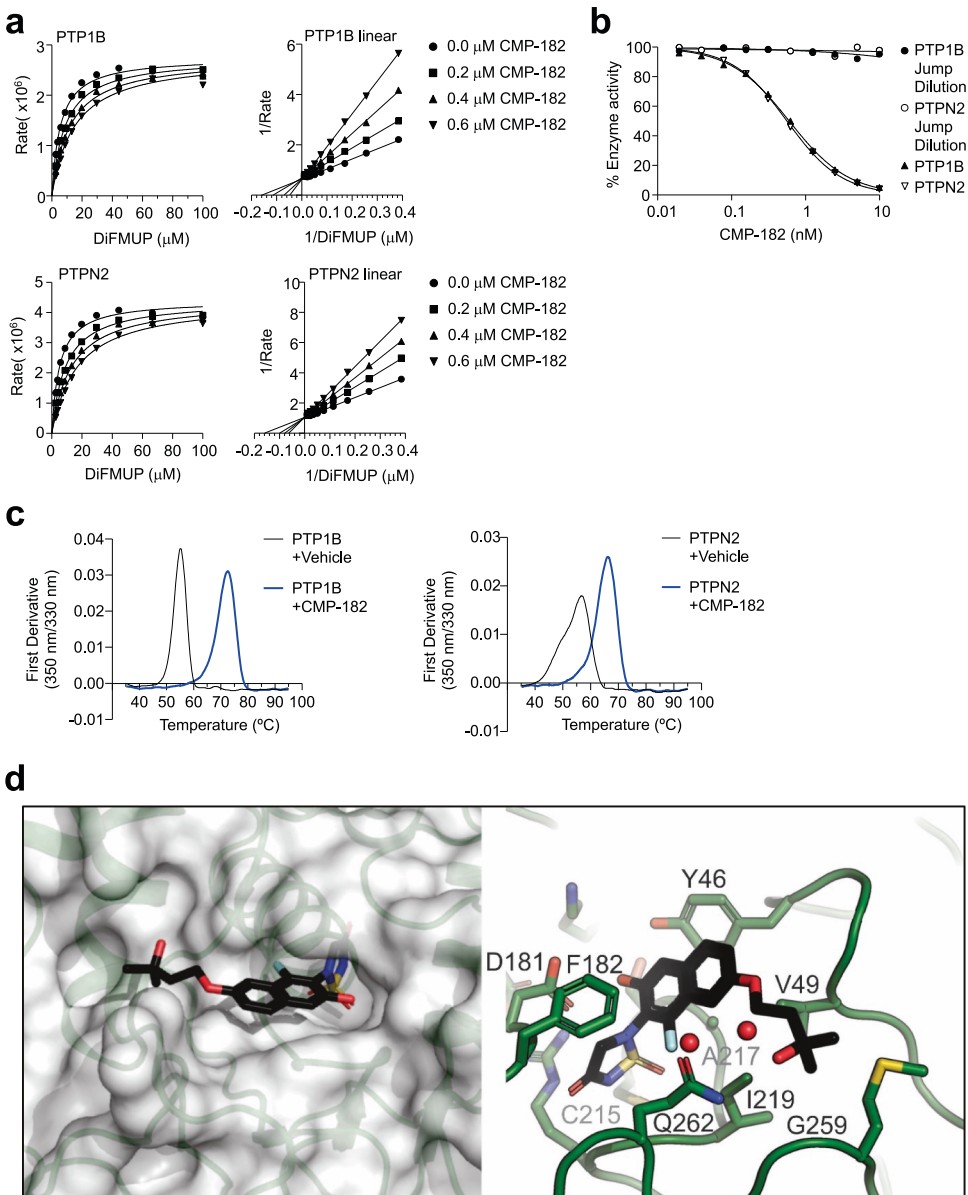

**Fig. 1 | Compound 182 is a reversible, competitive, active site inhibitor of PTP1B and PTPN2. a, b** Effect of Compound 182 (CMP-182) on PTP1B- and PTPN2-catalyzed 6,8-Difluoro-4-methylumbelliferyl phosphate (DiFMUP) hydrolysis. In (**a**) CMP-182 concentrations were 0 (●), 0.2 (■), 0.4(▲), and 0.6 nM (▼), respectively. **b** PTP1B- (●) and PTPN2- (○) catalyzed DiFMUP hydrolysis before and after jump dilution. **c** nanoDSF thermal shift first-derivative curves of 5 μM PTP1B or PTPN2 (black) and 5 μM PTP1B or PTPN2 + 50 μM CMP-182 (blue). **d** The structure of CMP-182 bound to PTP1B. Left, CMP-182 (stick representation) is shown bound to PTP1B

(semi-transparent surface). The thiadiazolidinone group mimics phosphotyrosine (pTyr) and is located deep in the active site, while the tail of CMP-182 is located in a shallow groove on the surface of the protein. Right, close-up view of CMP-182 in the active site. The thiadiazolidinone moiety lies above the catalytic Cys215. The naphthalene core of the compound sits on a hydrophobic surface of the protein with Phe182 from the WPD loop closed over the top. The methyl-butoxy tail is directed towards Gly259. In (**a–c**) representative results from two independent experiments are shown.

murine mammary tumor cells[53] into the inguinal mammary fat pads of female C57BL/6 mice and once tumors were established (20−30 mm²), administered Compound 182 intravenously at 10 mg/kg every 2 days or every 3 days and monitored for effects on tumor growth (Fig. 3a; Supplementary Fig. 8a). Compound 182 effectively repressed the growth of AT3-OVA mammary tumors (Fig. 3a) and this was accompanied by increased TILs (Fig. 3b, c), including increased CD44hiCD62Llo effector/memory and CD44hiCD62Lhi central memory CD4+ and CD8+ T cells (Fig. 3b; Supplementary Fig. 8b; Supplementary Fig. 9). The repression of tumor growth was more pronounced when Compound 182 was administered every two days (Fig. 3a; Supplementary Fig. 8a); hence for all subsequent experiments, mice were treated every two days. Infiltrating CD8+ T cells were more cytotoxic as

reflected by the proportion of cells staining for intracellular Granzyme B, TNF, and IFNγ and cell surface markers characteristic of activated effector T cells including, PD-1, Tim-3 and Lag-3 (Fig. 3d; Supplementary Fig. 8c); PD-1, Tim-3, and Lag-3 levels were not dramatically increased on a per cell basis (MFI), arguing against the promotion of T cell exhaustion otherwise associated with these markers (Supplementary Fig. 10a). By contrast neither splenic tissue weights, nor T cell numbers, including the proportion of activated and cytotoxic CD8+ T cells, were altered in the spleens of Compound 182-treated mice (Fig. Supplementary Fig. 10b, c). However, Compound 182 enhanced both basal p-SFK and p-STAT-5 in splenic CD8+ effector/memory T cells, as might be expected from inhibiting PTP1B and PTPN2[11,14] (Fig. 3e). Thus, these results indicate that Compound 182 can systemically inhibit

**Table 2 | Crystal structure data collection and refinement statistics**

| | PTP1B in complex with 182 |
|---|---|
| **Data collection** | |
| Wavelength (Å) | 0.9537 |
| Resolution range (Å) | 38.09 – 1.55 (1.605 – 1.55) |
| Space group | P $3_1$ 2 1 |
| Unit cell | |
| a, b, c (Å) | 87.96 87.96 104.34 |
| α, β, γ (°) | 90 90 120 |
| Total reflections | 136132 (13515) |
| Unique reflections | 68106 (6761) |
| Multiplicity | 2.0 (2.0) |
| Completeness (%) | 99.96 (100.00) |
| Mean I/sigma (I) | 16.37 (1.91) |
| Wilson B-factor (Å$^2$) | 22.61 |
| R-meas | 0.02314 (0.3715) |
| CC$_{1/2}$ | 1 (0.847) |
| CC* | 1 (0.958) |
| **Refinement** | |
| Reflections used in refinement | 68102 (6761) |
| Reflections used for R-free | 2585 (260) |
| R-work | 0.1747 (0.2407) |
| R-free | 0.1955 (0.2759) |
| Number of non-hydrogen atoms | 2735 |
| *Macromolecules* | 2455 |
| *Ligands* | 90 |
| *Solvent* | 233 |
| Protein residues | 299 |
| RMS (bonds) (Å) | 0.007 |
| RMS (angles) (°) | 0.91 |
| Ramachandran favored (%) | 97.64 |
| Ramachandran allowed (%) | 2.02 |
| Ramachandran outliers (%) | 0.34 |
| Rotamer outliers (%) | 1.11 |
| Overall molprobity score | 1.29 |
| Molprobity clash score | 3.93 |
| Average B-factor (Å$^2$) | 28.26 |
| *Macromolecules* | 27.29 |
| *Ligands* | 31.69 |
| *Solvent* | 37.75 |

Statistics for the highest-resolution shell are shown in parentheses.

PTP1B/PTPN2 in T cells, but this may not be sufficient to promote overt responses to endogenous antigens and promote systemic T cell expansion and activation. Instead, our findings point towards Compound 182 enhancing tumor antigen-induced T cell responses.

To further assess the therapeutic potential of targeting PTP1B and PTPN2 in cancer with small molecule inhibitors, we employed two additional syngeneic tumor models, MC38 colon and AT3 mammary tumor models. MC38 adenocarcinomas express high levels of MHC-I and recruit T cells[54], albeit not to the same extent as AT3-OVA tumors (Supplementary Fig. 10d). Moreover, unlike AT3-OVA tumors[11], MC38 tumors are unresponsive to PD-1 checkpoint blockade[54]. By contrast AT3 mammary tumors have comparatively few T cells (Supplementary Fig. 10d) and are also unresponsive to α-PD-1 therapy[10]. We found that Compound 182 repressed the growth of both subcutaneous MC38 colorectal tumors and orthotopic AT3 mammary tumors (Fig. 4a, d).

The repression of tumor growth was accompanied in each case by increased TILs (Fig. 4b, c, e, f). This included the increased recruitment of CD4$^+$ and CD8$^+$ effector/memory T cells (Fig. 4b, c, e, f). Tumor-infiltrating CD8$^+$ T cells were more activated (as reflected by the proportion of PD-1$^+$TIM-3$^+$ CD8$^+$ T cells) and exhibited enhanced cytotoxicity (as reflected by Granzyme B and IFNγ levels) (Fig. 4b, e). Taken together, our results demonstrate that targeting PTP1B and PTPN2 with Compound 182 represses the growth of immunologically cold and hot tumors, irrespective of whether they are responsive to PD-1 checkpoint blockade.

## T cell-dependent repression of AT3-OVA tumor growth

Beyond promoting the infiltration and/or expansion of central memory and effector/memory T cells and the accumulation of cytotoxic CD8$^+$ T cells in AT3-OVA mammary tumors, Compound 182 also increased the accumulation of natural killer (NK) cells and B cells that can promote anti-tumor immunity (Fig. 3c); this was also evident in MC38 and AT3 tumors (Fig. 4c, f). In AT3-OVA tumors the recruitment of NK cells and B cells was accompanied by decreased splenic NK and B cell numbers (Fig. 3c), in line with the accumulation of such cells within tumors reflecting infiltration from the periphery. In addition, Compound 182 also promoted the accumulation of CD4$^+$ regulatory T cells (T$_{regs}$) and myeloid-derived suppressor cells (MDSCs) that are immunosuppressive (Fig. 3c); again this was also evident for MC38 and AT3 tumors (Fig. 4c, f). To determine the extent to which this may reflect Compound 182 eliciting direct effects on such immune subsets, or otherwise being an outcome of T cell activation within tumors, we compared the effects of deleting either PTP1B or PTPN2 in T cells on AT3-OVA tumor growth, to the effects of Compound 182. To this end we implanted AT3-OVA tumor cells into the inguinal mammary fat pads of floxed control versus *Lck*-Cre;*Ptp1b*$^{fl/fl}$ or *Lck*-Cre;*Ptpn2*$^{fl/fl}$ T cell-specific PTP1B- or PTPN2-deficient female mice respectively and monitored tumor growth (Fig. 5a). As we have reported previously[11,14], the deletion of either PTP1B or PTPN2 in T cells was sufficient to repress the growth of AT3-OVA mammary tumors (Fig. 5a, b) and promote the recruitment of TILs, including CD4$^+$ and CD8$^+$ effector/memory T cells and activated CD8$^+$ T cells with enhanced cytotoxic potential (Fig. 5c), as reflected by the proportion of cells staining for the cell surface markers PD-1 and Tim-3 and those staining intracellularly for IFNγ, TNF and Granzyme B (Fig. 5d). The repression of tumor growth and the recruitment of T cells, including cytotoxic CD8$^+$ T cells, was generally more pronounced when PTPN2 was deleted in T cells (Fig. 5a, b), whereas the repression of tumor growth by Compound 182 more closely resembled the effects of deleting PTP1B in T cells (Fig. 5a, b). However, in each case, the deletion of PTP1B or PTPN2 in T cells, or the administration of Compound 182, were also accompanied by the increased recruitment of B cells and NK cells, as well as immunosuppressive T$_{regs}$ and MDSCs (Fig. 5e); once again NK cells, B cells and MDSCs were more pronounced when PTPN2 was deleted in T cells (Fig. 5e). Therefore, the deletion of PTP1B or PTPN2 in T cells for the most part phenocopies the effects of systemically administering Compound 182 on tumor growth and TILs.

To explore whether the Compound 182-mediated repression of tumor growth might reflect the inhibition of PTP1B/PTPN2 not only in T cells, but also in different immune subsets, we sought to determine if deleting PTPN2 throughout the entire immune system might elicit more pronounced anti-tumor immunity; we focussed on PTPN2 as the effects of deleting PTPN2 in T cells on immune cell recruitment were more pronounced when compared with the deletion of PTP1B in T cells. To delete PTPN2 in the hematopoietic compartment we took advantage of *Mx1*-Cre;*Ptpn2*$^{fl/fl}$ mice that we have described previously[25] in which PTPN2 can be inducibly deleted in all hematopoietic cells by the administration of double-stranded RNA poly (I:C)[25]. We implanted AT3-OVA tumor cells into the inguinal mammary fat pads of 8–10 week-old mice and administered poly (I:C) when large

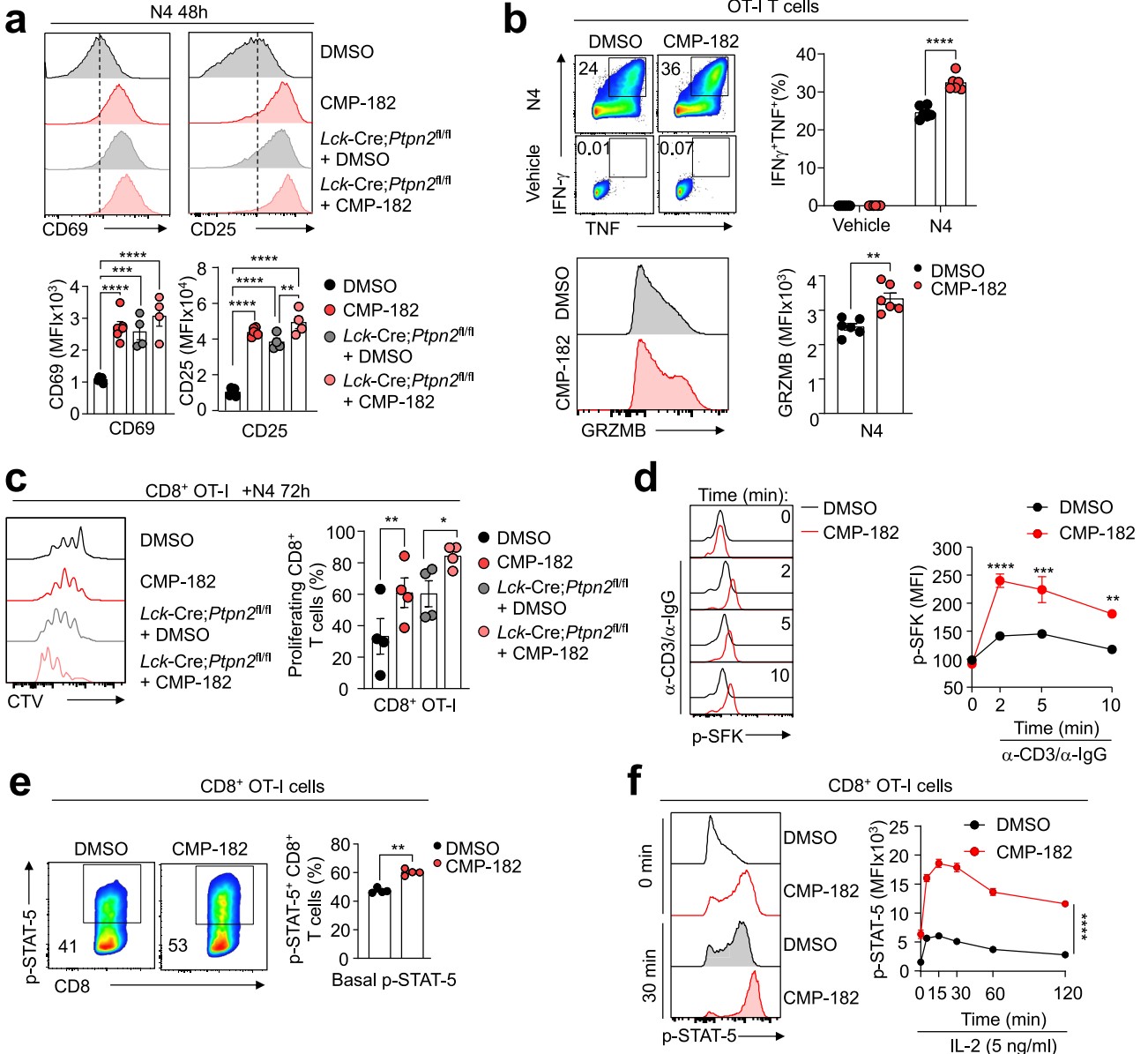

**Fig. 2 | Compound 182 promotes T cell signaling. a** OT-I cells isolated from wild type OT-I;*Ptpn2*$^{fl/fl}$ (*n* = 5–6) versus OT-I;*Lck*-Cre;*Ptpn2*$^{fl/fl}$ (*n* = 4) mice were stimulated with 1 nM of the cognate ovalbumin (OVA) peptide antigen SIINFEKL (N4) in the presence of vehicle (1% v/v DMSO) or Compound 182 (CMP-182; 1 μM) for 48 h and CD25 and CD69 mean fluorescent intensities (MFIs) in CD8$^+$ OT-I cells were determined by flow cytometry. **b** OT-I cells (*n* = 6) were activated with 1 nM N4 for 16 h and rested in media supplemented with IL-2 (5 ng/ml) and IL-7 (0.2 ng/ml) for 2 days. On day 3, T cells were pre-treated with vehicle or 1 μM CMP-182 overnight and stimulated with 1 nM N4 plus/minus 1 μM CMP-182 in the presence of Golgi-Plug™/GolgiStop™. IFN-γ$^+$, TNF$^+$ or Granzyme B$^+$ (GRZMB) CD8$^+$ T cells were analyzed by flow cytometry. **c** OT-I T cells isolated from wild-type OT-I; *Ptpn2*$^{fl/fl}$ (*n* = 4) versus OT-I;*Lck*-Cre; *Ptpn2*$^{fl/fl}$ (*n* = 4) mice were stained with CTV and stimulated with 0.5 nM N4 plus/minus 1 μM CMP-182 for 3 days and CTV dilution monitored by

flow cytometry. **d** Naive CD8$^+$ T cells (C57BL6) (*n* = 3) were preincubated with 10 μM CMP-182 for 4 h, the TCR cross-linked with α-mouse CD3ε (5 μg/ml) and goat anti-hamster IgG (20 μg/ml) and Y418 phosphorylated and activated SFK (p-SFK) MFIs determined by flow cytometry. **e** OT-I cells (C57BL/6) (*n* = 4) were stimulated with 1 nM N4 for 16 h, incubated plus/minus of 1 μM CMP-182 for 1 h and Y694 phosphorylated STAT-5 (p-STAT-5) in CD8$^+$ OT-I cells assessed by flow cytometry. **f** OT-I cells (C57BL/6) (*n* = 6) were stimulated with 1 nM N4 for 16 h, incubated plus/minus 1 μM CMP-182 for 1 h, and then stimulated with IL-2 (5 ng/ml) plus/minus 1 μM CMP-182 and p-STAT-5 MFIs in CD8$^+$ OT-I cells determined by flow cytometry. In (**a**–**f**) representative results (means ± SEM) from at least two independent experiments are shown. Significances in (**a**, **c**) were determined using a 1-way ANOVA Test and in (**b**, **e**) using a 2-tailed Mann-Whitney U Test. In (**d**, **f**) significances were determined using a 2-way ANOVA Test.

tumors (40–50 mm²) were established (Fig. 6a). The deletion of PTPN2 in the hematopoietic compartment repressed tumor growth and increased TILs, but this was not more pronounced than that associated with the deletion of PTP1B or PTPN2 in T cells (Fig. 6a, b; Fig. 5). Interestingly, tumor-infiltrating NK cells were reduced (Fig. 6b), probably as a consequence of the overt inflammation and autoimmunity that otherwise accompanies the deletion of PTPN2 in the hematopoietic system[25].

Given that the administration of Compound 182 to tumor-bearing mice was accompanied by the pronounced recruitment of NK cells, which can have both direct cytotoxic effects, as well indirect antitumor effects through IFNγ-dependent T cell recruitment[55], we also assessed if Compound 182 might at least in part act by inhibiting PTP1B or PTPN2 in NK cells. To this end we deleted either PTP1B or PTPN2 in NK cells (Supplementary Fig. 11a, b) using the *Ncr1*-Cre transgene that deletes exclusively in NK cells[56]. The deletion of either PTP1B or PTPN2

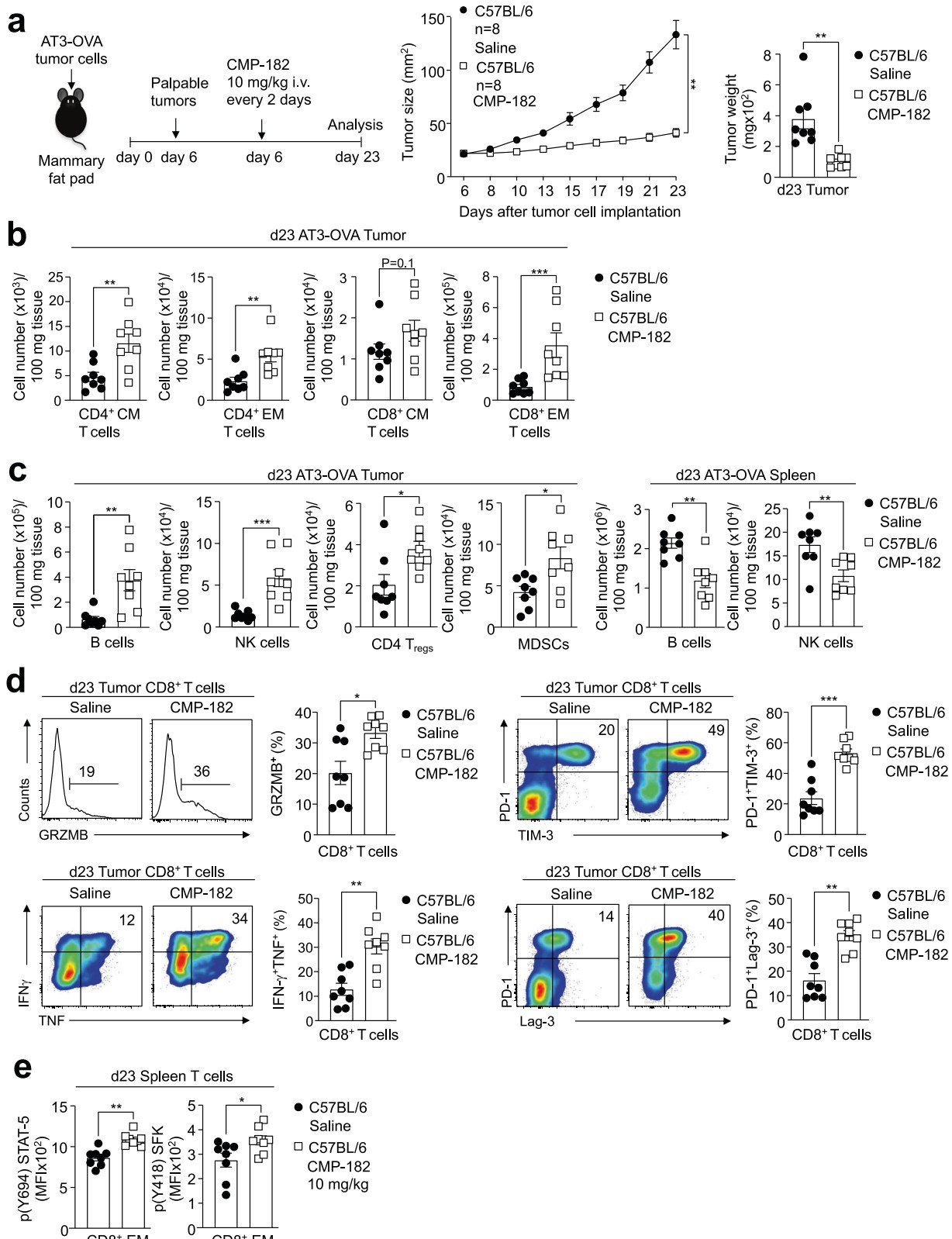

had no significant effect on the NK cell development or frequency in the periphery (Supplementary Fig. 11c, d). To assess the impact of deleting either PTP1B or PTPN2 in NK cells on anti-tumor immunity we implanted $B2m^{-/-}$ MC38 adenocarcinoma cells lacking MHC-I [sensitive to NK cells but not T cells[57]] (Supplementary Fig. 11e, f) or a 70:30 mix of wild type MC38s (insensitive to NK cells) and $B2m^{-/-}$ MC38s (Fig. 6c, d) into the flanks of floxed control versus $Ncr1$-Cre;$Ptpn1^{fl/fl}$ or

$Ncr1$-Cre;$Ptpn2^{fl/fl}$ male mice deficient for PTP1B or PTPN2 respectively in NK cells. We found that the deletion of either PTP1B or PTPN2 in NK cells had no significant impact on tumor growth and mouse survival (Supplementary Fig. 11e-f; Fig. 6c, d). Consistent with this, Compound 182 did not enhance IL-15-induced p-STAT-5 or NK cell proliferation (CTV dilution) (Supplementary Fig. 12a, b), nor did it affect NK cell activation (as assessed by the proportion of IFNγ or CD107A

**Fig. 3 | Compound 182 represses AT3-OVA mammary tumor growth.** AT3-OVA mammary tumor cells were injected into the fourth inguinal mammary fat pads of 8-week-old C57BL/6 female mice. Mice were treated with Compound 182 (CMP-182; 10 mg/kg i.v.; $n = 8$) or saline ($n = 8$) on days (d) 6, 8, 10, 12, 14, 16, 18 and 21 after tumor cell implantation. **a** Tumor growth was monitored and tumor weights measured. **b, c** Tumor-infiltrating lymphocytes or splenocytes including CD44$^{hi}$CD62L$^{hi}$ CD8$^+$ and CD4$^+$ central memory (CM) T cells, CD44$^{hi}$CD62L$^{lo}$ CD8$^+$ and CD4$^+$ effector/memory (EM) T cells, CD19$^+$ B cells, NK1.1$^+$TCRβ$^-$ (NK) cells, CD4$^+$CD25$^+$FoxP3$^+$ regulatory T cells (T$_{regs}$) and granulocytic and monocytic CD11b$^+$F4/80$^{hi/lo}$Ly6C$^+$Ly6G$^{+/-}$ myeloid-derived suppressor cells (MDSCs) were analyzed by flow cytometry. **d** Tumor-infiltrating T cells from (**a**) were stimulated with PMA/Ionomycin in the presence of Golgi Stop/Plug and stained for intracellular IFN-γ and TNF. Intracellular granzyme B (GRZMB), surface PD-1, TIM-3, and Lag-3 were detected in unstimulated tumor-infiltrating CD8$^+$ T cells. **e** p-STAT-5 and p-SFK MFIs were assessed in splenic CD8 + EM T cells. In (**a–e**) representative results (means ± SEM) from at least two independent experiments are shown. Significance for tumor sizes in (**a**) was determined using a 2-way ANOVA Test and for tumor weights in (**a**) using a 2-tailed Mann-Whitney U Test. In (**b–e**) significances were determined using a 2-tailed Mann−Whitney U Test.

expressing cells) in response to cytokines or by crosslinking activation receptors (NK1.1, Ly49H) (Supplementary Fig. 12c, d). These results suggest that the anti-tumor activity of Compound 182 cannot be attributed to promotion of NK cell-dependent anti-tumor immunity.

To specifically determine the extent to which the induction of anti-tumor activity by Compound 182 may be reliant on targeting PTP1B and/or PTPN2 in T cells, we next asked if Compound 182 could repress the growth of AT3-OVA tumors in *Rag1$^{-/-}$* (C57BL/6) mice that lack T cells and B cells. We found that Compound 182 had no effect on AT3-OVA mammary tumor growth in immunodeficient *Rag1$^{-/-}$* mice (Fig. 6e). Although we cannot exclude contributions from B cells, these results are consistent with the repression of tumor growth being attributed to the inhibition of PTP1B and/or PTPN2 in T cells. To explore this further, we also asked if Compound 182 could enhance the repression of tumor growth otherwise achieved by deleting PTPN2 in T cells. Although the effects were modest, Compound 182 administration additionally repressed the growth of AT3-OVA tumors in *Lck*-Cre;*Ptpn2$^{fl/fl}$* mice (Fig. 7a, b). However, this was not accompanied by any significant changes in TILs, including CD4$^+$ and CD8$^+$ effector/memory T cells, B cells, NK cells, T$_{regs}$ or MDSCs when compared to the corresponding TILs in *Lck*-Cre;*Ptpn2$^{fl/fl}$* mice (Fig. 7c). Instead, CD8$^+$ T cell cytotoxicity, as reflected by IFNγ$^+$TNF$^+$ infiltrating CD8$^+$ T cells, was significantly increased in mice treated with Compound 182 (Fig. 7d). Since Compound 182 also enhanced the TCR-induced activation and proliferation of *Lck*-Cre;*Ptpn2$^{fl/fl}$* CD8$^+$ T cells ex vivo (Fig. 2a, c) we surmise that the enhanced repression of tumor growth may be attributed to the inhibition of PTP1B in PTPN2-deficient T cells. Irrespective, taken together our findings are consistent with Compound 182 acting predominantly on T cells to facilitate anti-tumor immunity and repress the growth of immunogenic tumors.

## Targeting PTPN2 in T cells promotes AT3-OVA tumor inflammation, STAT-1 signaling and T cell recruitment

Our studies indicate that the deletion of PTPN2 in T cells or the administration of Compound 182 similarly recruits TILs and promotes T cell activation in immunogenic AT3-OVA tumors to repress tumor growth. Previous studies, including our own, have shown that the deletion of PTPN2 in poorly immunogenic tumors, such as B16F10A or AT3 syngeneic tumors, can significantly enhance T cell recruitment and anti-tumor immunity by promoting IFN-induced STAT-1 Y701 phosphorylation (p-STAT-1) and the expression of chemokines such as CXCL9 and CXCL10 in tumor cells[6,10] to facilitate T cell and NK cell recruitment. By contrast, the deletion of PTPN2 in aggressive E0771 mammary tumors which exhibit heightened IFNγ/IFNβ expression, STAT-1 signaling, and T cell infiltrates independent of PTPN2 status, as well as a robust immunosuppressive tumor microenvironment, had no effect on tumor growth[10]. Similarly, in this study we found that the deletion of PTPN2 in immunogenic AT3-OVA tumors using CRISPR ribonucleoprotein (RNP)-based genome editing (Supplementary Fig. 13) had no significant effect on tumor growth (Supplementary Fig. 14). Therefore, the effects of Compound 182 on TILs and anti-tumor immunity in immunogenic AT3-OVA mammary tumors, are unlikely to be mediated by direct effects on tumor cells.

Next, we asked if the inhibitor's effects on TILs and anti-tumor immunity in AT3-OVA mammary tumors might nonetheless occur as a consequence of enhanced intratumoral T cell activation and resultant inflammatory STAT-1 signaling in tumors. To this end we compared the effects of Compound 182 with the deletion of PTPN2 in T cells on the promotion of p-STAT-1 and the recruitment of T cells. We found that the deletion of PTPN2 in T cells resulted in robust p-STAT-1 within the nuclei of AT3-OVA tumor cells; both the number of p-STAT-1 positive tumor cells and p-STAT-1 intensity were increased by the deletion of PTPN2 in T cells (Fig. 7e). This, in turn, was accompanied by the increased expression of STAT-1 target genes within tumors, including those encoding the T cell/NK cell chemoattractant CXCL9 (encoded by *Cxcl9*) and inhibitory ligands for immune checkpoints, including programmed death-ligand 1 (PD-L1; *Cd274*) and major histocompatibility complex II (MHC-II; *H2ab1*) (Fig. 7f). Although other STAT-1 target genes, including antigen presentation genes (*Tapb*, *Tap1*, and *H2k1*), were not increased, we did find that IFNγ (*Ifng*) mRNA levels were significantly increased (Fig. 7f), probably reflecting the accumulation of IFNγ producing cytotoxic CD8$^+$ T cells and NK cells (Fig. 3c, d). Indeed, immunohistochemical assessment reaffirmed the increased infiltration of CD3ε$^+$ T cells (Fig. 7e). As with the deletion of PTPN2 in T cells, we found that the systemic administration of Compound 182 and the repression of AT3-OVA tumor growth also increased p-STAT-1 staining within the tumor, as well as the abundance of CD3ε$^+$ T cells (Fig. 7g).

Having established that the deletion of PTPN2 in T cells is sufficient to promote STAT-1 signaling and CD3ε T cell infiltration and repress AT3-OVA tumor growth, we next determined if administering the inhibitor might be accompanied by further increases in intratumoral STAT-1 signaling and T cell infiltration (Fig. 7e). We found that the administration of Compound 182 to AT3-OVA tumor-bearing *Lck*-Cre;*Ptpn2$^{fl/fl}$* mice did not further increase p-STAT-1 signaling (Fig. 7e, f) or T cell infiltrates (as assessed by immunohistochemistry or flow cytometry) (Fig. 7c, e), indicating that any direct effects on tumor cells and/or stromal cells are unlikely to significantly affect STAT-1 inflammatory responses and T cell recruitment. Taken together our findings indicate that at least in immunogenic tumors, Compound 182 may be efficacious not only by inhibiting PTP1B/PTPN2 in T cells to drive T cell activation, expansion and cytotoxicity, but also by promoting T cell-mediated inflammation and consequent STAT-1 signaling in tumor cells to exacerbate T cell recruitment and anti-tumor immunity.

## T cell-dependent and -independent repression of AT3 tumor growth

The deletion of PTPN2 in B16F10A melanoma or AT3 mammary tumors in C57BL/6 mice can promote IFN-induced STAT-1 signaling and the expression of STAT-1 target genes, including those encoding chemokines such as CXCL9 and antigen presentation genes such as MHC-I (*H2k1*) to facilitate T cell-mediated anti-tumor immunity[6,10]. We noted that although the deletion of PTPN2 in AT3-OVA tumors did not significantly affect tumor growth (Figs. S13–14), the deletion of PTPN2 in AT3 cells markedly repressed tumor growth (Fig. 8a; Supplementary Fig. 13) and this was accompanied by a marked increase in TILs, including effector T cells and NK cells and the expression of STAT-1

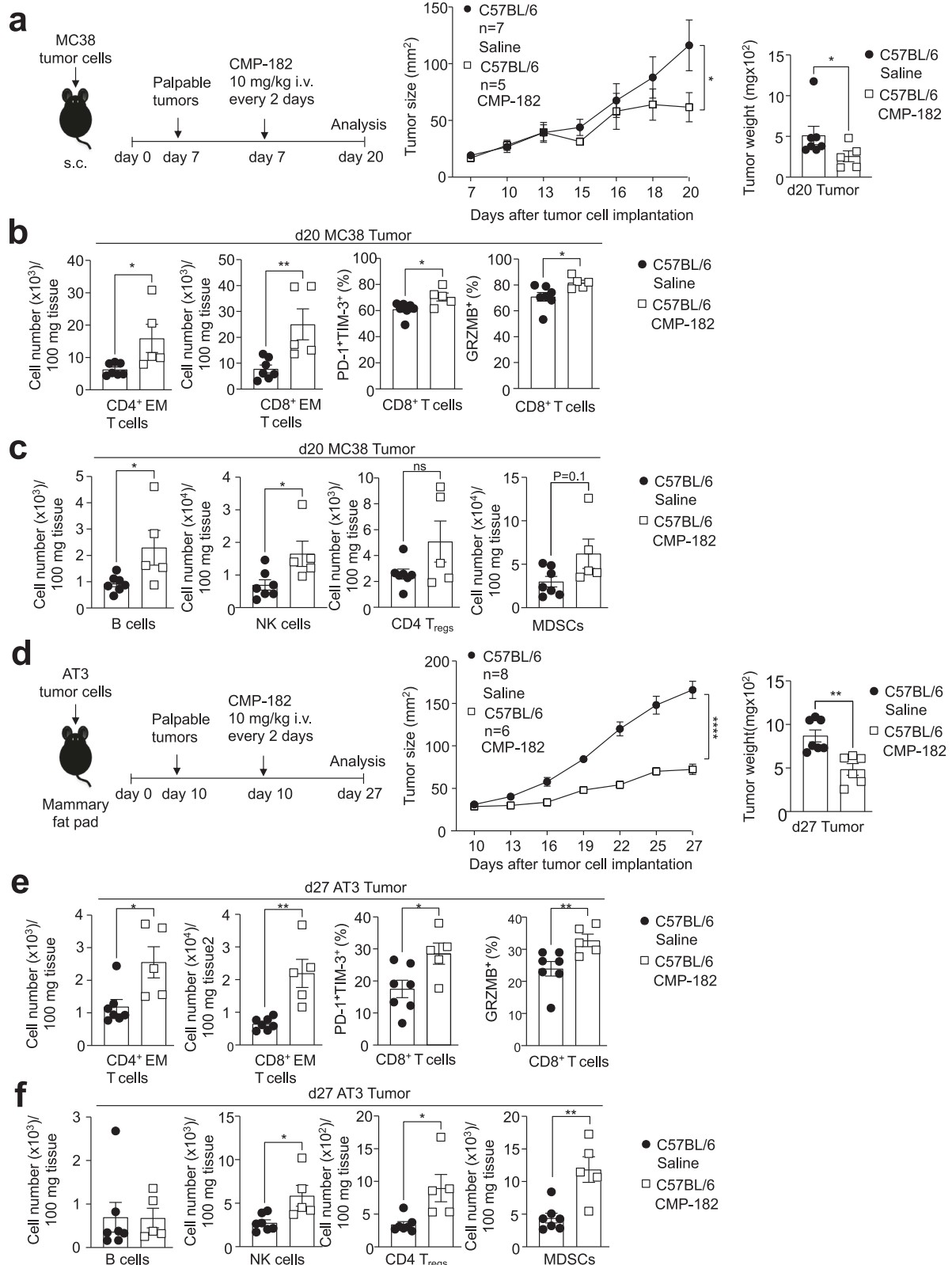

target genes, including *Cxcl9*, *Cxcl10*, *Cd274* and *H2k1* (Fig. 8b, c). Thus, we reasoned that in immunologically cold AT3 tumors, Compound 182 might repress tumor growth by eliciting effects both on tumor cells and recruited T cells. We have shown previously that the combined deletion of PTPN2 in AT3 tumor cells and T cells results in a greater repression of tumor growth than deleting PTPN2 either in tumor cells or T cells[10]. Accordingly, we determined if Compound 182 could

repress the growth of AT3 tumors beyond that achieved by deleting PTPN2 in T cells. We implanted AT3 tumor cells into the inguinal mammary fat pads of *Ptpn2*^fl/fl control versus *Lck*-Cre;*Ptpn2*^fl/fl T cell-specific PTPN2-deficient female mice and once tumors were established (20-30 mm²) treated tumor-bearing mice with Compound 182. As reported previously[10] we found that the deletion of PTPN2 in T cells alone was sufficient to significantly repress AT3 tumor growth

**Fig. 4 | Compound 182 represses MC38 and AT3 tumor growth. a–c** MC38 colon tumor cells were xenografted into the flanks of 8-week-old C57BL/6 male mice. Mice were treated with Compound 182 (CMP-182; 10 mg/kg i.v.; $n = 5$) or saline ($n = 7$) on days (d) 7, 9, 11, 13, 15, 17, and 19 after tumor cell implantation. **a** Tumor growth was monitored and tumor weights were measured. **b, c** Tumor-infiltrating lymphocytes (TILs) including CD44$^{hi}$CD62L$^{lo}$ CD8$^+$ and CD4$^+$ effector/memory (EM) T cells, CD19$^+$ B cells, NK1.1$^+$TCRβ$^-$ (NK) cells, CD4$^+$CD25$^+$FoxP3$^+$ regulatory T cells (T$_{regs}$) and granulocytic and monocytic CD11b$^+$F4/80$^{hi/lo}$Ly6C$^+$Ly6G$^{+/-}$ myeloid-derived suppressor cells (MDSCs) were analyzed by flow cytometry. In (**b**) intracellular granzyme B (GRZMB) and cell surface PD-1 and TIM-3 were detected in unstimulated tumor-infiltrating CD8$^+$ T cells. **d–f** AT3 mammary tumor cells were injected into the fourth inguinal mammary fat pads of 8-week-old C57BL/6 female mice. Mice were treated with CMP-182 (10 mg/kg i.v.; $n = 6$) or saline ($n = 8$) on days (d) 10, 12, 14, 16, 18, 20, 22, 24, and 26 after tumor cell implantation. **d** Tumor growth was monitored and tumor weights measured. **e, f** TILs (CMP-182: $n = 5$; Saline: $n = 7$) including CD4$^+$ EM T cells, CD19$^+$ B cells, NK cells, T$_{regs}$ and MDSCs were analyzed by flow cytometry. In (**e**) intracellular GRZMB and cell surface PD-1 and TIM-3 were detected in unstimulated tumor-infiltrating CD8$^+$ T cells. In (**a–f**) representative results (means ± SEM) from at least two independent experiments are shown. Significance for tumor sizes in (**a, d**) was determined using a 2-way ANOVA Test and for tumor weights in (**a, d**) using a 2-tailed Mann–Whitney U Test. In (**b, c, e, f**) significances were determined using a 2-tailed Mann–Whitney U Test.

(Fig. 8d). Moreover, the combined deletion of PTPN2 in T cells and treatment with Compound 182 further suppressed tumor growth and/or led to the eradication of 3/6 tumors (Fig. 8d). The marked repression of tumor growth was accompanied by the enhanced promotion of tumor p-STAT-1 and CD3$^+$ T cell infiltrates (Fig. 8e). Taken together, these results suggest that in immunologically cold tumors, Compound 182 may elicit synergistic effects on anti-tumor immunity by targeting PTP1B/PTPN2 both in tumor cells and T cells.

### Compound 182 sensitizes AT3 tumors to PD-1 checkpoint blockade

Antibodies blocking PD-1 result in durable clinical responses in immunogenic tumors that have abundant T-cell infiltrates[4]. However, not all tumors with TILs are responsive to such therapy and resistance is common[4]. Previous studies have established that the deletion of PTPN2 in immunologically cold tumors sensitizes tumors to PD-1 blockade by promoting the IFN-induced and STAT-1-mediated expression of antigen presentation genes, T cell recruitment as well as the expression of ligands for immune checkpoints, including PD-L1[6,10]. Our recent studies have shown that PTPN2 deletion sensitizes otherwise therapy-resistant AT3 mammary tumors to PD-1 blockade[10] whereas in this study we have shown that Compound 182 can increase TILs and repress AT3 tumor growth (Fig. 4d–f). The enhanced repression of tumor growth was accompanied by increased tumor STAT-1 signaling, as reflected by the increased expression of STAT-1 target genes, including *Cd274* (encodes PD-L1) (Fig. 9a). Accordingly, we assessed the impact of Compound 182 on the response of AT3 tumors to α-PD-1 therapy (Fig. 9b–f). We found that Compound 182 not only repressed the growth of AT3 tumors, but also rendered otherwise largely resistant AT3 tumors sensitive to PD-1 checkpoint blockade (Fig. 9b, c). Indeed, the combination therapy led to a marked repression of tumor growth that was accompanied by the synergistic recruitment of CD4$^+$ and CD8$^+$ effector/memory T cells and CD8$^+$ T cells with enhanced cytotoxic potential (Fig. 9d, e); the combination therapy also led to synergistic increases in other TILs including NK cells (Fig. 9f). Therefore, our studies indicate that pharmacologically targeting PTP1B and PTPN2 with Compound 182 might not only recruit and activate T cells to repress the growth of immunologically cold tumors, but also sensitize otherwise resistant tumors to PD-1 checkpoint blockade to synergistically promote anti-tumor immunity.

### Compound 182 does not promote systemic inflammation or autoimmunity

The targeting of immune checkpoints such as PD-1 on the surface of T cells has proven effective in the treatment of many cancers, but adverse immune-related events, including cytokine release syndrome (CRS) and autoimmunity, are common[58]. Since PTP1B and PTPN2 can function as intracellular checkpoints to tune T cell responses[7,11,14], one possible adverse consequence of systemically targeting PTP1B and PTPN2 with small molecule inhibitors may be the development of CRS and autoimmunity. This may be especially pertinent for PTPN2, since *PTPN2* loss of function SNPs have been associated with autoimmunity[41–44]. Accordingly, we assessed the impact of systemically

targeting PTP1B and PTPN2 with Compound 182 in tumor-bearing mice on the development of systemic inflammation and autoimmunity (Fig. 10). Although Compound 182 effectively repressed AT3-OVA mammary tumor growth, this was not accompanied by overt signs of morbidity or systemic inflammation, since spleen weights and splenic T cell and myeloid numbers were unaltered (Supplementary Fig. 10b, c), and serum levels of pro-inflammatory cytokines, including IL-6, IFNγ, and TNF were not significantly increased (Fig. 10a). In addition, we did not observe any overt lymphocytic infiltrates in non-lymphoid tissues, including liver, lungs, salivary glands and colon, as might otherwise be expected with the development of CRS (Fig. 10b). Moreover, there were no clear signs of autoimmunity, since serum anti-nuclear antibodies (ANA) were unaltered and there were no signs of tissue damage (Fig. 10c). Although circulating levels of the liver enzyme aspartate aminotransferase (AST) were moderately increased, serum alanine aminotransferase (ALT) levels were unaffected (Fig. 10c) and there were no signs of liver fibrosis, as assessed by monitoring for collagen deposition by histology (Picrosirius red) (Fig. 10b). By contrast, the repression of tumor growth in *Mx1*-Cre;*Ptpn2*$^{fl/fl}$ mice in which PTPN2 was deleted throughout the hematopoietic compartment was accompanied by inflammation and overt morbidity. Tumor-bearing poly (I:C)-treated *Mx1*-Cre;*Ptpn2*$^{fl/fl}$ mice developed dermatitis (Fig. 10d), had splenomegaly associated with the expansion of T cells and myeloid cells (Fig. 10e), had systemic inflammation as reflected by increased serum levels of the proinflammatory cytokines IL-6, IFNγ and TNF (Fig. 10f) and developed overt liver damage as reflected by the increased circulating levels of ALT and AST (Fig. 10g) and the presence of lymphocytic infiltrates and fibrosis (Fig. 10h). Taken together our findings demonstrate that systemic targeting of PTP1B and PTPN2 with a small molecule inhibitor can effectively repress tumor growth without necessarily promoting CRS and autoimmunity.

## Discussion

Considerable progress has been made in developing drugs targeting different members of the PTP superfamily for a variety of human indications, including various cancers, type 2 diabetes, diabetic macular edema, and Rett syndrome[59, 60]. Early attempts to design selective inhibitors of the prototypic family member PTP1B for metabolic disease were hampered by issues of specificity due to the conserved nature of the PTP active site[59]. In particular, attempts to develop active site inhibitors with the capacity to differentiate PTP1B over the closely related PTPN2, initially failed[59]. This challenge has now been overcome with the advent of allosteric inhibitors that bind at sites away from the PTP active site[47,61,62]. Nonetheless, recent studies have defined the roles of PTP1B and PTPN2 in the immune system and their potential as targets for cancer immunotherapy and somewhat paradoxically, this has sparked interest in the development of drugs capable of targeting both PTP1B and PTPN2[6,7,10,11,14,21]. In this study, we have explored the therapeutic efficacy and safety of a small molecule active site competitive inhibitor of PTP1B and PTPN2, Compound 182. Our studies demonstrate that Compound 182 is a highly specific equipotent inhibitor of PTP1B and PTPN2 that can markedly repress the growth of syngeneic tumors in mice and enhance the response to PD-1

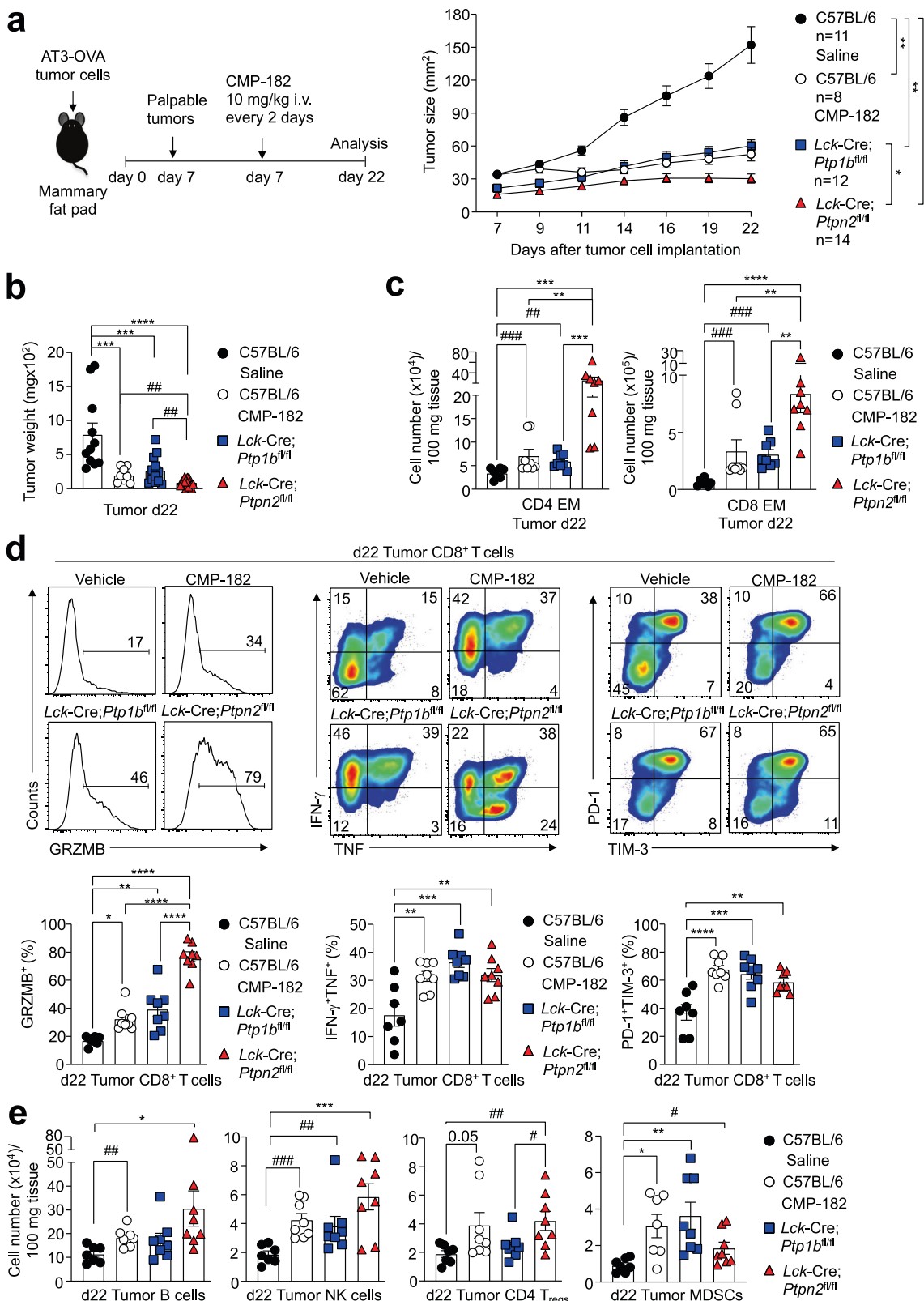

checkpoint blockade without promoting overt immune-related toxicities.

In this study we focused on Compound 182 since the structures of AbbVie's clinical candidates ABBV-CLS-484 and ABBV-CLS-579 were not revealed when our work began and Compound 182 was one of the best AbbVie/Calico inhibitors for PTP1B and PTPN2 based on information in the disclosed patent (WO2019246513A1). Both Compound 182 and ABBV-CLS-484 contain the acylsulfamide pTyr mimetic, which as expected, bound to the conserved PTP active site. Nonetheless, despite the PTP active site being highly conserved across all classical pTyr phosphatases, Compound 182 exhibited remarkable sub-nanomolar potency, with orders of magnitude greater selectivity for PTP1B and PTPN2 over other PTPs tested. This was attributed to the combination of three residues, Gly259, Ala217 and Val49, which are

**Fig. 5 | Comparable anti-tumor immunity induced by Compound 182 and the deletion of PTP1B or PTPN2 in T cells.** AT3-OVA mammary tumor cells were injected into the fourth inguinal mammary fat pads of 8-week-old C57BL/6 ($n = 8$–11 in each group), *Lck*-Cre;*Ptp1b*$^{fl/fl}$ (C57BL/6) ($n = 12$) and *Lck*-Cre;*Ptpn2*$^{fl/fl}$ (C57BL/6) ($n = 14$) female mice. Mice were treated with Compound 182 (CMP-182; 10 mg/kg i.v.; $n = 8$) or saline ($n = 11$) on days (d) 7, 9, 11, 13, 15, 17, 19, and 21 after tumor cell implantation. **a** Tumor growth was monitored and **b** tumor weights were measured. **c**–**e** Tumor-infiltrating lymphocytes (TILs) (CMP-182: $n = 8$; Saline: $n = 7$; *Lck*-Cre;*Ptp1b*$^{fl/fl}$: $n = 8$; *Lck*-Cre;*Ptpn2*$^{fl/fl}$: $n = 8$) including CD44$^{hi}$CD62L$^{lo}$ CD8$^+$ and CD4$^+$ effector/memory (EM) T cells, CD19$^+$ B cells, NK1.1$^+$TCRβ$^-$ (NK) cells, CD4$^+$CD25$^+$FoxP3$^+$ regulatory T cells (T$_{regs}$) and granulocytic and monocytic CD11b$^+$F4/80$^{hi/lo}$Ly6C$^+$Ly6G$^{+/-}$ myeloid-derived suppressor cells (MDSCs) were analyzed by flow cytometry. In (**d**) Tumor-infiltrating T cells were stimulated with PMA/Ionomycin in the presence of Golgi Stop/Plug and stained for intracellular IFN-γ and TNF. Intracellular granzyme B (GRZMB), surface PD-1 and TIM-3 were detected in unstimulated tumor-infiltrating CD8$^+$ T cells. In (**a**–**e**) representative results (means ± SEM) from at least two independent experiments are shown. Significance for tumor sizes in (**a**) was determined using a 2-way ANOVA Test and for tumor weights in (**b**) and for TILs in (**c**–**e**) using a 1-way ANOVA Test. In (**b, c, e**) significances were determined using a 2-tailed Mann–Whitney U Test ($^#p < 0.05$, $^{##}p < 0.01$, $^{###}p < 0.001$) where indicated.

unique to PTP1B and PTPN2, with all other classical pTyr phosphatases having residues with bulkier side chains that would sterically hinder binding of the terminal methyl group, the naphthalene core and the ether group of Compound 182, respectively. Based on structure we predict that ABBV-CLS-484 would bind similarly and potentially have comparable specificity, with the only major differences being in the central core (the benzocyclohexane moiety in ABBV-CLS-484 is not planar) and the end of the tail of Compound 182 (the *tert*-butanol group of Compound 182 is replaced by a less bulky isobutane). However, given that bicyclic naphthyl pTyr mimetics are known to bind stronger to the PTP active site than the corresponding single ring compounds[63,64], it is possible that Compound 182 may even have a higher affinity for PTP1B and PTPN2 than ABBV-CLS-484. Indeed, when assayed using DiFMUP as a substrate, Compound 182 inhibited PTP1B and PTPN2 with IC$_{50}$ values of $0.63 ± 0.01$ and $0.58 ± 0.02$ nM, respectively (Supplementary Fig. 15). Under the same conditions, the IC$_{50}$ values of ABBV-CLS-484 for PTP1B and PTPN2 are $1.84 ± 0.09$ and $1.60 ± 0.06$ nM respectively (Supplementary Fig. 15), approximately 3-fold lower than those of Compound 182. Nonetheless, the extent to which the two compounds would ultimately inhibit PTP1B and PTPN2 in vivo may be very different, as this would be influenced by additional factors including cell permeability and pharmacokinetics.

In this study, we found that Compound 182 effectively repressed the growth of immunogenic tumors, including AT3-OVA mammary tumors and MC38 colorectal tumors that were predominated by TILs, as well as orthotopic AT3 mammary tumors that had comparatively few T cell infiltrates and were resistant to α-PD-1 therapy. The repression of tumor growth in each case was accompanied not only by the recruitment and activation of effector T cells, but also by the infiltration of additional lymphocytes, including B cells and NK cells, as well as immunosuppressive T$_{regs}$ and MDSCs. Although we cannot exclude the possibility that Compound 182 might elicit direct effects in multiple immune subsets to influence tumor growth, its ability at least to repress the growth of AT3-OVA mammary tumors was attributed to the recruitment and activation of T cells, as no overt effect was evident in *Rag1*$^{-/-}$ mice that lack T cells. However, the administration of Compound 182 also increased the recruitment for example of NK cells that can kill tumor cells directly, as well as facilitate the recruitment of cytotoxic T cells[55]. Although neither PTP1B nor PTPN2 deletion affected NK development or tumor growth in vivo, and the inhibitor did not affect NK cell activation in vitro, it is likely that the increased abundance of NK cells and consequent increased IFNγ production contribute to the therapeutic efficacy of Compound 182 in vivo.

Previous studies have shown that the deletion of PTPN2 in tumor cells can enhance anti-tumor immunity by promoting IFN/JAK/STAT-1 signaling to drive T cell recruitment and MHC-I-dependent antigen presentation[6,10]. Our studies indicate that the relative importance of PTP1B/PTPN2 inhibition in tumor cells versus T cells may be dependent largely on the tumor microenvironment. For immunogenic tumors with abundant T cell infiltrates, such as AT3-OVA mammary tumors, Compound 182 might largely repress tumor growth by acting directly on T cells, since the effects of the inhibitor could be largely phenocopied by deleting either PTP1B or PTPN2 in T cells and this was not

further enhanced by the administration of Compound 182. The deletion of PTPN2 in T cells not only promoted the activation and cytotoxic potential of tumor-infiltrating/resident T cells (as reflected by IFNγ, TNF and GZMB levels), but also enhanced STAT-1 signaling in AT3-OVA tumor cells, the expression of *Cxcl9* and the recruitment of T cells. The deletion of PTP1B or PTPN2 in T cells was also accompanied by the recruitment of B cells, NK cells and immunosuppressive cells. Therefore, the inhibition/deletion of PTP1B and/or PTPN2 in resident/infiltrated T cells in immunogenic tumors would not only facilitate T cell activation, but also, as a consequence of heightened CD8$^+$ T cell IFNγ production in the tumor microenvironment, exacerbate inflammation and promote the further recruitment and activation of T cells to repress tumor growth. Indeed, in T cell-specific PTPN2-deficient mice, Compound 182 had no significant additional effect on STAT-1 signaling and TILs and only moderately enhanced T cell cytotoxicity and the repression of AT3-OVA tumor growth. By contrast, Compound 182 significantly repressed the growth AT3 tumors and enhanced the repression otherwise achieved by deleting PTPN2 in T cells. The repression of AT3 tumor growth was accompanied by increases in STAT-1 signaling, the expression of STAT-1 target genes, such as *Cxcl9* and *Cd274*, the recruitment of T cells, and the re-sensitization of otherwise resistant AT3 tumors to PD-1 immunotherapy. Therefore, Compound 182 can elicit both direct effects on tumor cells and T cells to facilitate T cell recruitment and activation respectively in immunologically cold tumors, or otherwise activate T cells in immunogenic tumors to drive inflammatory STAT-1 signaling and exacerbate T cell recruitment/activation and anti-tumor immunity. Although in the longer term, tumor-intrinsic resistance to PTP1B/PTPN2 inhibition may eventuate, the beneficial anti-tumor effects of PTP1B/PTPN2 inhibition in T cells would nonetheless persist. Moreover, although persistent T cell activation might normally be accompanied by the development of T cell exhaustion, this might also be overcome by the inhibition of PTP1B/PTPN2, since the deletion of PTPN2 and the promotion IFNAR/STAT-1 signaling in T cells has been shown to promote the expansion of progenitor T cells, to not only replenish the exhausted effector pool, but also sustain responses to PD-1 blockade[7,11,65].

Although in this study we have focused primarily on the actions of PTP1B and PTPN2 in T cells and tumor cells, PTP1B and PTPN2 also have important roles in other immune subsets, which may variably influence the therapeutic efficacy of Compound 182. For example, both PTP1B and PTPN2 have been implicated in dendritic cell (DC) maturation and antigen presentation and thereby T cell priming/activation[8,9,66,67]. However, although the partial inactivation of PTP1B and/or PTPN2 may promote the development of immunogenic DCs and anti-tumor immunity, the complete loss of either PTP can impair DC maturation and function[9,66,67]. Therefore, the extent to which the inhibition of PTP1B and/or PTPN2 in DCs with Compound 182 might ultimately impact on tumor growth remains unclear. Furthermore, it is important to recognize that in some circumstances, the inhibition of PTP1B and/or PTPN2 in tumor cells might instead contribute to tumor growth. This potential is highlighted by studies demonstrating that *PTPN2* deletion in T-cell acute lymphoblastic leukemias in humans is associated with oncogenic JAK/STAT signaling[68], whereas in mice the

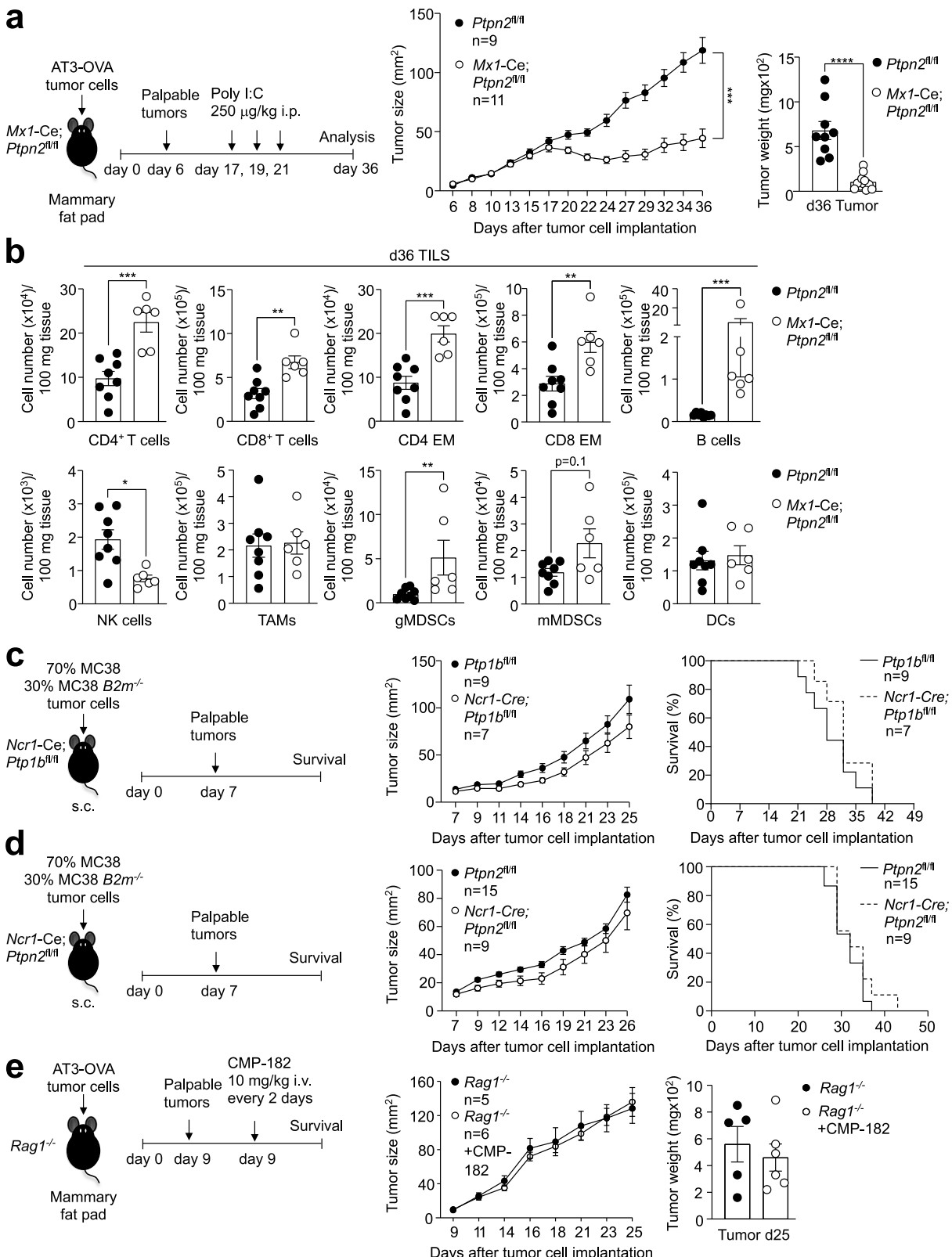

**a**

**b** d36 TILS

**c**

**d**

**e**

deletion of PTPN2 can also promote STAT-3 signaling to facilitate the development of hepatocellular carcinomas[33] or skin carcinogenesis[69]. Similarly, the myeloid-specific homozygous deletion of PTP1B can promote STAT-3 signaling and the development of acute myeloid leukemia[70]. Nonetheless, perturbations in the IFNγ/JAK/STAT-1 response are responsible for the ability of many tumors to develop resistance to immunotherapy[4,5]. Moreover, this pathway is more

broadly critical to anti-tumor immunity, including T cell/CAR T cell cellular therapies[4,71,72]. Therefore, it is likely that the pharmacological targeting of two key negative regulators of this pathway, PTP1B and PTPN2, would for the most part be beneficial and help promote anti-tumor immunity and the response to immunotherapy.

The development of immune-related toxicities is a common and often limiting complication of immune-targeted therapies[58]. As

**Fig. 6 | The effects of Compound 182 on anti-tumor immunity are reliant on T cells but not NK cells.** AT3-OVA mammary tumor cells were injected into the fourth inguinal mammary fat pads of 8-week-old *Mx1*-Cre;*Ptpn2*fl/fl (C57BL/6) female mice (*n* = 9-11 in each group). Mice were treated with poly I:C (250 µg/kg i.v.) to inducibly delete PTPN2 on days (d) 17, 19, and 21 after tumor cell implantation. **a** Tumor growth was monitored and tumor weights were measured. **b** Tumor-infiltrating lymphocytes (TILs) (*Ptpn2*fl/fl: *n* = 8; *Mx1*-Cre;*Ptpn2*fl/fl: *n* = 6) including CD4+ and CD8+ T cells, CD44hiCD62Llo CD8+ and CD4+ effector/memory (EM) T cells, CD19+ B cells, NK1.1+TCRβ− (NK) cells, CD11b+F4/80hiLy6C−Ly6G− tumor-associated macrophages (TAMs), granulocytic CD11b+F4/80hi/loLy6CintLy6G+ (gMDSCs) and monocytic CD11b+F4/80hi/lo Ly6C+Ly6G− (mMDSCs) myeloid-derived suppressor cells and CD11c+ (DCs) dendritic cells were analyzed by flow cytometry. **c, d** Wild type MC38 (70%) and *B2m*−/− MC38 (30%) tumor cells were xenografted into the flanks (subcutaneous; s.c.) of either 8-week-old **c**) *Ncr1*-Cre;*Ptp1b*fl/fl (C57BL/6) or **d**) 8 week-old *Ncr1*-Cre;*Ptpn2*fl/fl (C57BL/6) male mice or the corresponding floxed control mice and tumor growth and survival (*Ptp1b*fl/fl: *n* = 9; *Ncr*-Cre;*Ptp1b*fl/fl: *n* = 7; *Ptpn2*fl/fl: *n* = 15; *Ncr1*-Cre;*Ptpn2*fl/fl: *n* = 9) monitored. **e** AT3-OVA mammary tumor cells were injected into the fourth inguinal mammary fat pads of 8-week-old *Rag1*−/− (C57BL/6) female mice. Mice were treated with Compound 182 (CMP-182; 10 mg/kg i.v.; *n* = 6) or saline (*n* = 6) on days 9, 11, 13, 15, 17, 19, and 21 after tumor cell implantation and tumor growth was monitored. In (**a**−**e**) representative results (means ± SEM) from at least two independent experiments are shown. Significance for tumor sizes in (**a, c**−**e**) was determined using a 2-way ANOVA Test and for tumor weights in (**a**) using a 2-tailed Mann–Whitney U Test. In (**b**) significance was determined using a 2-tailed Mann–Whitney U Test.

evident when targeting other important immunomodulatory molecules, such as PD-1[58], deficiencies in PTPN2 in mice and humans are accompanied by systemic inflammation, overt T cell auto-reactivity, and morbidity[21,23,25,34,35,41–44]. In our studies, we found that the intravenous administration of Compound 182 every other day effectively repressed tumor growth by recruiting T cells, however, it neither promoted systemic inflammation nor autoimmunity, at least not in the timeframe examined. Phase I trials are currently underway assessing the safety and efficacy of the PTP1B/PTPN2 inhibitors in patients with locally advanced or metastatic tumors (NCT04417465, NCT04777994). It is possible that in a subset of individuals, especially those prone to autoimmune disease, the administration of such PTP1B/PTPN2 inhibitors may result in immune-related toxicities, as also seen in patients receiving immune checkpoint therapies[58]. Nonetheless, our preclinical studies point toward the existence of a therapeutic window for safely and effectively targeting PTPN2 and/or PTP1B and enhancing anti-tumor immunity and the response to immunotherapy.

## Methods

All protocols were approved by the Monash University School of Biomedical Sciences Animal Ethics Committee (Ethics numbers: 23177, 36697) and by the Monash University Institutional Biosafety Committee (NLRD Identifier PC2-N05/14).

### Synthetic experimental methods

Chemicals and solvents were purchased from standard suppliers and no further purifications were required. Deuterated solvents were purchased from Cambridge Isotope Laboratories, Inc. $^{1}$H and $^{13}$C NMR spectra were recorded on a BRUKER Avance III Nanobay 400 MHz NMR spectrometer equipped with a BACS 60 automatic sample changer at 400 MHz and 101 MHz, respectively. Chemical shifts were reported in parts per million (ppm) and all peaks were referenced through the residual deuterated solvent peak. Multiplicity was indicated as followed: s (singlet); d (doublet); t (triplet); q (quartet); m (multiplet); dd (doublet of doublet); br s (broad singlet). The coupling constants were reported in Hz.

Thin-layer chromatography analysis (TLC) was performed on precoated silica gel aluminium-backed plates. Visualization was done by using either stains such as ninhydrin or under UV light at 254 and 365 nm. Flash column chromatography was run using P60 silica gel (40-63 µm).

Low resolution mass spectrometry was obtained by Agilent 1260 Infinity II LCMS SQ equipped with a 1260 Infinity G1312B Binary pump and a 1260 Infinity G1367E 1260 HiP ALS autosampler. Detection of UV reactive compounds was performed at wavelengths of 214 nm and 254 nm and was recorded by a 1290 Infinity G4212iA 1290 DAD variable wavelength detector. LC-MS data was processed through the LC/MSD Chemstation Rev.B.04.03 SP2 coupled with MassHunter Easy Access Software. The LC component was run as a reverse phase HPLC using a Raptor C18 2.7 µm 50 × 3.0 mm column at 35 °C. The following buffers were used: Buffer A: 0.1% formic acid in water; buffer B: 0.1% formic acid in MeCN. The following gradient was used with a Poroshell 120 EC-C18 3.0 × 50 mm 2.7-micron column with a flow rate of 0.5 mL/min and a total run time of 5 minutes: 0- 2 minutes 5–100% buffer B; 2–4.5 minutes 100% buffer B; 4.5–5 minutes 100–5% buffer B. Mass spectra were in positive and negative ion mode with a scan range of 100–1000 *m/z*. UV detection was run at 214 and 254 nm. The retention times ($t_{R}$) are in minutes.

High resolution mass spectrometry was obtained by Agilent 6224 TOF LC/MS Mass Spectrometer coupled to an Agilent 1290 Infinity (Agilent, Palo Alto, CA). All data were acquired and reference mass corrected via a dual-spray electrospray ionization (ESI) source. Each scan or data point on the Total Ion Chromatogram (TIC) is an average of 13,700 transients, producing a spectrum every second. Mass spectra were created by averaging the scans across each peak and background subtracted against the first 10 seconds of the TIC. Acquisition was performed using the Agilent Mass Hunter Data Acquisition software version B.05.00 Build 5.0.5042.2 and analysis was performed using Mass Hunter Qualitative Analysis version B.05.00 Build 5.0.519.13. The mass spectrometer drying gas flow was at 11 L/min at a temperature of 325 °C in electrospray ionization mode. The nebulizer was setup at 45 psi with a capillary voltage of 4000 V. The fragmentor, skimmer and OCT RFV voltage were 160 V, 65 V and 750 V, respectively. The scan range acquired were 100–1500 *m/z*.

Analytical high-performance liquid chromatography (HPLC) was performed on Agilent 1260 Analytical HPLC with a 1260 DAD: G4212B detector and a Zorbax Eclipse Plus C18 Rapid Resolution 4.6 × 100 mm 3.5-Micron column. The eluent system was made up of solvent A (H$_2$O with 0.1% formic acid) and solvent B (MeCN with 0.1% formic acid). Samples used the same method: gradient starts from 95% solvent A and 5% solvent B and reaches 100% solvent B in 8 min, sustained at 100% solvent B for 1 min, returned to 95% solvent A and 5% solvent B over 0.1 min and sustained at 95% solvent A and 5% solvent B for 0.9 min.

### Synthesis of Compound 182
#### 3-(Benzyloxy)-7-bromo-2-naphthoic acid (II).

A suspension of 7-bromo-3-hydroxy-2-naphthoic acid (5 g, 18.72 mmol, 1 equiv.) and Cs$_2$CO$_3$ (12.20 g, 37.44 mmol, 2 equiv.)

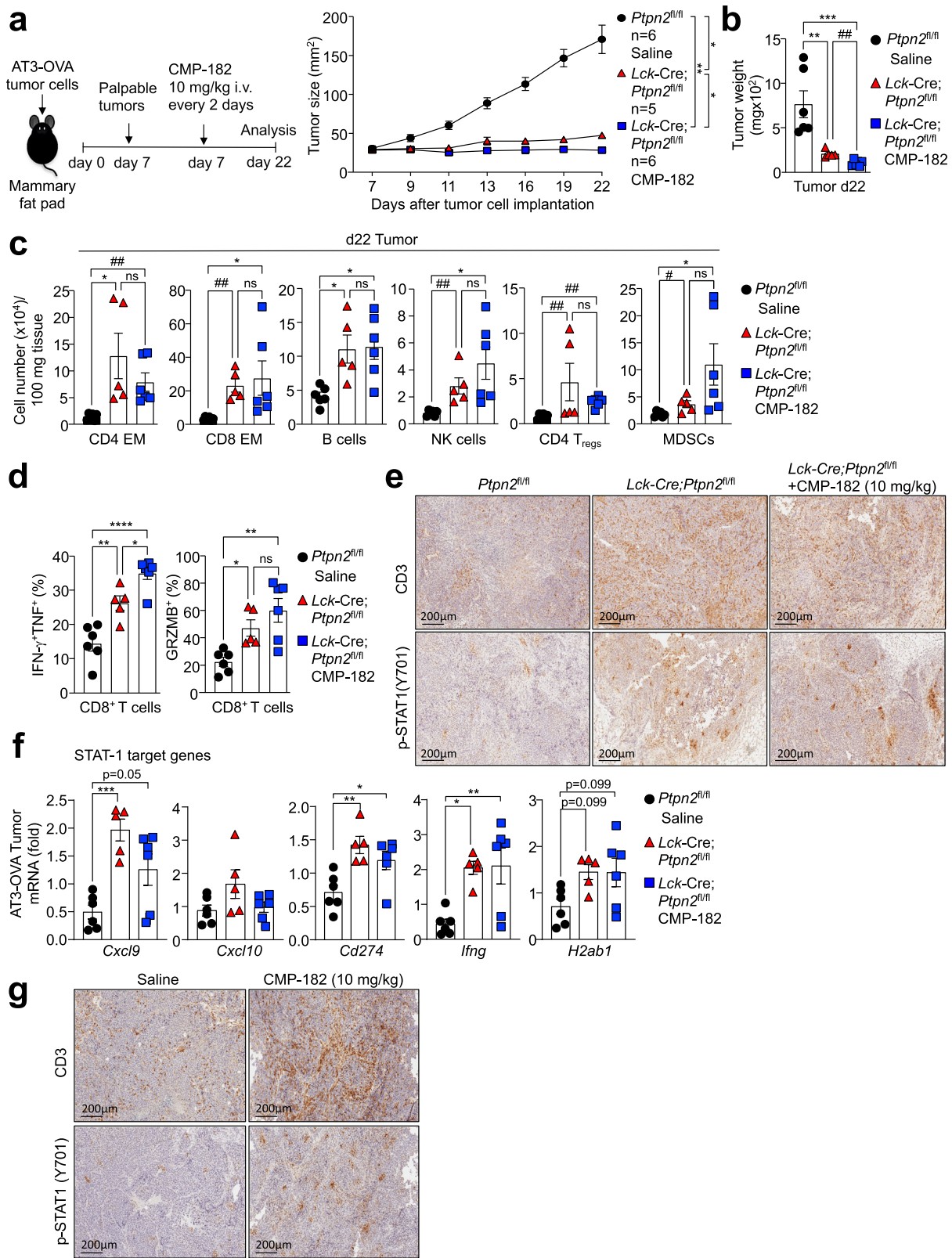

in DMF (35 mL) was stirred rapidly for 5 min at room temperature. Benzyl bromide (4.45 mL, 37.44 mmol, 2 equiv.) was added to the mixture and stirred rapidly for 2 h. Once starting material was consumed, $H_2O$ (100 mL) was poured into the yellow suspension and stirred for 5 min. The resulting precipitate was filtered under

vacuum, washing with $H_2O$ (50 mL) and 1:2 *tert*-butyl methyl ether/cyclohexane (40 mL). The yellow solid and LiOH (535.5 mg, 22.35 mmol, 2 equiv.) were suspended in 1:2 $H_2O$ / MeOH (150 mL) and heated to 80 °C. After 16 h stirring, the mixture was cooled to room temperature, acidified with 2 M HCl aq. solution (20 mL,

**Fig. 7 | Effects of Compound 182 on AT3-OVA mammary tumor growth, STAT-1 signaling and T cell infiltrates in *Lck*-Cre;*Ptpn2*^fl/fl^ mice. a–f** AT3-OVA mammary tumor cells were injected into the fourth inguinal mammary fat pads of 8 week-old *Ptpn2*^fl/fl^ (*n* = 6) and *Lck*-Cre;*Ptpn2*^fl/fl^ (C57BL/6) (*n* = 5-6 in each group) female mice. Mice were treated with Compound 182 (CMP-182; 10 mg/kg i.v.; *n* = 6) or saline (*n* = 6) on days (d) 7, 9, 11, 13, 15, 17, 19 and 21 after tumor cell implantation. **a** Tumor growth was monitored and **b** tumor weights were measured. **c** Tumor-infiltrating lymphocytes including CD44^hi^CD62L^lo^ CD8^+^ and CD4^+^ effector/memory (EM) T cells, CD19^+^ B cells, NK1.1^+^TCRβ^−^ (NK) cells, CD4^+^CD25^+^FoxP3^+^ regulatory T cells (T_regs_) and granulocytic and monocytic CD11b^+^F4/80^hi/lo^Ly6C^+^Ly6G^+/−^myeloid-derived suppressor cells (MDSCs) were analyzed by flow cytometry. **d** Tumor-infiltrating T cells from (**a**) were stimulated with PMA/Ionomycin in the presence of Golgi Stop/Plug and stained for intracellular IFN-γ and TNF. Intracellular granzyme B (GRZMB) was detected in unstimulated tumor-infiltrating CD8^+^ T cells. AT3-OVA tumors were processed for **e** immunohistochemistry staining for p-STAT-1, or CD3

(counterstained with hematoxylin) or **f** qPCR monitoring for the expression of STAT-1 target genes. **g** AT3-OVA mammary tumor cells were injected into the fourth inguinal mammary fat pads of C57BL/6 mice. Mice were treated with CMP-182 (10 mg/kg i.v.) or saline on days 6, 8, 10, 12, 14, 16, 18, and 21 after tumor cell implantation; the resultant tumor growth curves are shown in Fig. 4a. The resulting AT3-OVA tumors were processed for immunohistochemistry staining for STAT-1 Y701 phosphorylation (p-STAT-1), or CD3 (counterstained with hematoxylin). In (**a**–**d**, **f**) representative results (means ± SEM) from at least two independent experiments are shown. In (**e**, **g**) micrographs are representative of two independent experiments with 5 mice per group. Significance for tumor sizes in (**a**) was determined using a 2-way ANOVA Test and for tumor weights in (**b**) using a 1-way ANOVA Test. In (**c**, **d**) significances were determined using a 1-way ANOVA Test and a 2-tailed Mann–Whitney U Test (^#^$p < 0.05$, ^##^$p < 0.01$) where indicated. In (**f**) significances were determined using a 1-way ANOVA Test.

40 mmol) and stirred rapidly for 15 min. The resulting white solid was filtered under vacuum, washed with $H_2O$ (100 mL) then *tert*-butyl methyl ether (30 mL), and dried to obtain the product as an off-white solid (6.30 g, 94% yield over two steps). ^1^H NMR (400 MHz, DMSO-$d_6$) δ 8.16 (d, *J* = 2.0 Hz, 1H), 8.01 (s, 1H), 7.77 (d, *J* = 8.8 Hz, 1H), 7.59 (dd, *J* = 8.7, 2.1 Hz, 1H), 7.57 – 7.52 (m, 2H), 7.50 (s, 1H), 7.44 – 7.37 (m, 2H), 7.36 – 7.29 (m, 1H), 5.25 (s, 2H). ^13^C{^1^H} NMR (101 MHz, DMSO-$d_6$) δ 167.8, 154.0, 136.9, 133.0, 129.9, 129.7, 128.8, 128.6, 128.3, 128.1, 127.6, 127.2, 116.7, 107.8, 69.4. One quaternary carbon signal was not observed. HRMS (ESI-TOF) *m/z*: calcd for $C_{18}H_{13}BrO_3Na$ [M^79^Br + Na]^+^ 378.9940, found 378.9958; [M^81^Br + Na]^+^ 380.9922, found 380.9918.

### 3-(Benzyloxy)−7-bromonaphthalen-2-amine (III).

To a suspension of compound **II** (6.33 g, 17.64 mmol, 1 equiv.) in 100 mL toluene/*tert*-butanol (1:1) was added triethylamine (5.69 mL, 26.46 mmol, 1.2 equiv.). The white suspension was heated to 80 °C under $N_2$ atmosphere, and diphenylphosphoryl azide (3.61 mL, 26.46 mmol, 1.5 equiv.) was added dropwise to the mixture over 10 min. After heating at 80 °C overnight, the mixture was cooled to room temperature, diluted with $H_2O$ (100 mL), and extracted with EtOAc (2 × 50 mL). The combined organic layers were washed with brine (2 × 30 mL), dried over $Na_2SO_4$, filtered, and concentrated to an off-white solid. The solid was suspended in 30 mL diethylenetriamine under $N_2$ atmosphere and heated to 130 °C. After heating for 16 h, the mixture was allowed to cool to room temperature before $H_2O$ (100 mL) was added slowly to the dark brown mixture and resulted in yellow precipitation and heat evolution. Once cooled, the mixture was extracted with $CH_2Cl_2$ (3 × 100 mL). The combined organic layer was washed with brine (2 × 50 mL), dried over $Na_2SO_4$, filtered, and concentrated to a solid. The solid was triturated with isopropanol (20 mL) to form a slurry and filtered to obtain the product as an off-white solid (2.35 g, 40% yield over two steps). ^1^H NMR (400 MHz, DMSO-$d_6$) δ 7.70 (d, *J* = 2.0 Hz, 1H), 7.58 – 7.51 (m, 3H), 7.45 – 7.38 (m, 2H), 7.37 – 7.31 (m, 1H), 7.29 (s, 1H), 7.18 (dd, *J* = 8.6, 2.0 Hz, 1H), 6.89 (s, 1H), 5.25 (s, 2H). ^13^C{^1^H} NMR (101 MHz, DMSO-$d_6$) δ 147.3, 139.7, 136.8, 131.5, 128.43, 128.40, 127.8, 127.5, 126.0, 125.3, 124.0, 116.6, 106.5, 105.4, 69.3. HRMS (ESI-TOF) *m/z*: calcd for $C_{17}H_{14}BrNO$

[M^79^Br + H]^+^ 328.0332, found 328.0331; [M^81^Br + H]^+^ 330.0313, found 330.0311.

### Methyl (3-(benzyloxy)−7-bromonaphthalen-2-yl)glycinate (IV).

To a vigorously stirred suspension of **III** (1.70 g, 5.18 mmol, 1 equiv.) and $K_2CO_3$ (1.43 g, 10.36 mmol, 2 equiv.) in 50 mL DMF was added methyl 2-bromoacetate (0.735 mL, 7.77 mmol, 1.5 equiv.) and the mixture was heated to 60 °C and monitored until completion by LC-MS (typically after 16 h). The mixture was extracted between $H_2O$ (50 mL) and EtOAc (100 mL). The aqueous layer was back-extracted with EtOAc (2 × 50 mL). The combined organic layer was washed with $H_2O$ (5 × 50 mL), brine (2 × 50 mL), dried over $Na_2SO_4$, filtered, and concentrated to a brown solid. The product was obtained after flash chromatography on silica gel (eluent 0–20% EtOAc / petroleum ether) as an off-white solid (1.59 g, 77% yield). ^1^H NMR (400 MHz, CDCl$_3$) δ 7.74 (s, 1H), 7.52 – 7.36 (m, 6H), 7.27 (d, *J* = 8.2 Hz, 1H), 7.07 (s, 1H), 6.57 (s, 1H), 5.22 (s, 2H), 4.04 (d, *J* = 5.4 Hz, 2H), 3.81 (s, 3H). ^13^C{^1^H} NMR (101 MHz, CDCl$_3$) δ 171.2, 147.5, 138.4, 136.5, 131.7, 128.8, 128.4, 128.2, 127.7, 127.5, 126.1, 125.8, 118.0, 106.3, 102.9, 70.6, 52.5, 45.3. HRMS (ESI-TOF) *m/z*: calcd for $C_{20}H_{18}BrNO_3$ [M^79^Br + H]^+^ 400.0543, found 400.0550; [M^81^Br + H]^+^ 402.0525, found 402.0533.

### Methyl *N*-(3-(benzyloxy)−7-bromo-1-fluoronaphthalen-2-yl)-*N*-sulfamoylglycinate (V).

To a brown solution of **IV** (2.10 g, 5.50 mmol, 1 equiv.) in 80 mL THF was added *N*-fluorobenzenesulfonimide (2.08 g, 6.60 mmol, 1.2 equiv.) and the mixture was stirred for 6 h. The

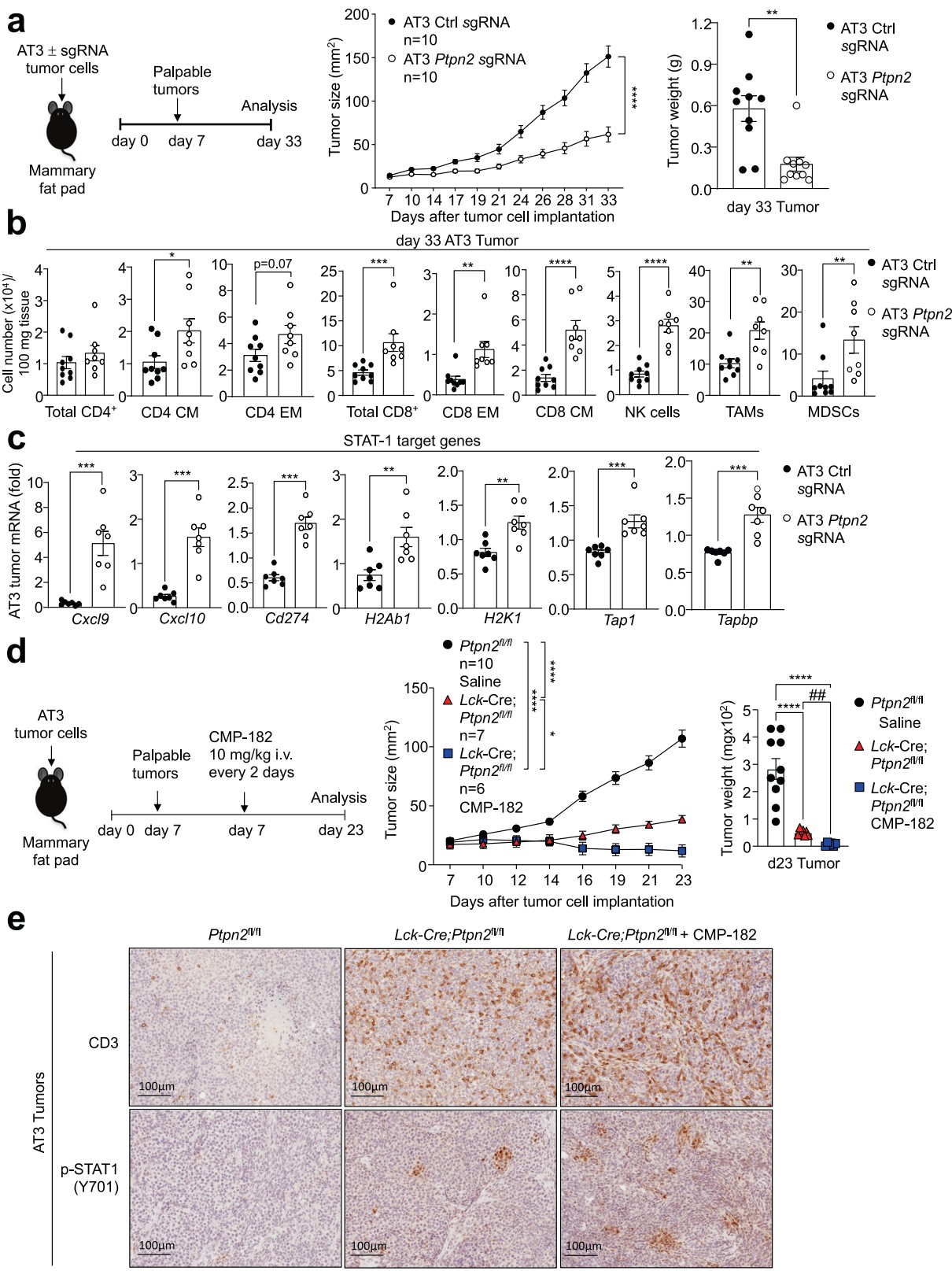

residual oxidant was quenched with sodium thiosulfate pentahydrate (2.00 g, 5.69 mmol, 1 equiv.) in 10 mL $H_2O$ and the mixture was stirred for 30 min. The mixture was diluted with 10 mL $H_2O$, and extracted with 60 mL EtOAc. The organic layer was washed with sat. $NaHCO_3$ aq. solution (2 × 30 mL), brine (2 × 30 mL), dried over $Na_2SO_4$, filtered and concentrated to an orange solid. The product was obtained after flash chromatography on silica gel (eluent 0 – 20% EtOAc / petroleum ether) as a pale yellow solid (1.42 g, 62% yield). $^1H$ NMR (400 MHz, DMSO-$d_6$) δ 7.80 (s, 1H), 7.66 (d, $J$ = 8.7 Hz, 1H), 7.55 (d, $J$ = 7.1 Hz, 2H), 7.46 – 7.40 (m, 2H), 7.40 – 7.33 (m, 2H), 7.30 (s, 1H), 5.30 (s, 2H), 4.22 (d, $J$ = 4.0 Hz, 2H), 3.63 (s, 3H). $^{19}F$ NMR (377 MHz, DMSO-$d_6$) δ −149.4. HRMS

**Fig. 8 | Compound 182 promotes STAT-1 signaling and increases T cell infiltrates in AT3 tumors. a** AT3 control cells (Ctl sgRNA: *n* = 10) or those in which PTPN2 had been deleted by CRISPR RNP (*Ptpn2* sgRNA: *n* = 10) were injected into the fourth inguinal mammary fat pads of 8-week-old female C57BL/6 mice and tumor growth was monitored. **b** Tumor-infiltrating lymphocytes (Ctl sgRNA: *n* = 9; *Ptpn2* sgRNA: *n* = 8) including total CD4+ and CD8+ T cells, CD44hiCD62Lhi CD8+ and CD4+ central memory (CM) T cells, CD44hiCD62Llo CD8+ and CD4+ effector/memory (EM) T cells, NK1.1+TCRβ− (NK) cells, CD11b+F4/80hiLy6C−Ly6G− tumor-associated macrophages (TAMs) and granulocytic and monocytic CD11b+F4/80hi/loLy6C+Ly6G+/− myeloid-derived suppressor cells (MDSCs) were analyzed by flow cytometry. **c** AT3 tumors were processed for qPCR monitoring for the expression of STAT-1 target genes (Ctl sgRNA: *n* = 7; *Ptpn2* sgRNA: *n* = 7). **d, e** AT3 mammary tumor cells were injected into the fourth inguinal mammary fat pads of 8 week-old *Ptpn2*fl/fl (*n* = 10) and *Lck*-Cre;*Ptpn2*fl/fl (*n* = 6–7 per group) (C57BL/6) female mice. Mice were treated with CMP-182 (10 mg/kg i.v.; *n* = 6) or saline (*n* = 7) on days 7, 9, 11, 13, 15, 17, 19 and 21 after tumor cell implantation. **d** Tumor growth was monitored and tumor weights were measured. **e** The resulting AT3 tumors were processed for immunohistochemistry staining for p-STAT-1 (Y701) and CD3 (counterstained with hematoxylin). In (**a**–**d**) representative results (means ± SEM) from at least two independent experiments are shown. In (**e**) micrographs are representative from two independent experiments with 5 mice per group. Significance for tumor sizes in (**a, d**) was determined using a 2-way ANOVA Test and for tumor weights in a) using a 2-tailed Mann−Whitney U Test and (**d**) using a 1-way ANOVA Test. In (b-c and where indicated by ## *p* < 0.01) significances were determined using a 2-tailed Mann−Whitney U Test.

(ESI-TOF) *m/z*: calcd for $C_{20}H_{17}BrFNO_3$ $[M^{79}Br + H]^+$ 418.0449, found 418.0459; $[M^{81}Br + H]^+$ 420.0431, found 420.0442.

## Methyl *N*-(3-(benzyloxy)−7-bromo-1-fluoronaphthalen-2-yl)-*N*-sulfamoylglycinate (VII).

To a solution of chlorosulfonyl isocyanate (0.205 mL, 2.36 mmol, 1.5 equiv.) in anhydrous $CH_2Cl_2$ (6 mL) at 0 °C was added *tert*-butanol (0.226 mL, 2.36 mmol, 1.5 equiv.) dropwise. After the mixture was stirred at 0 °C for 1 h, a preformed mixture of compound **V** (658.2 mg, 1.57 mmol, 1 equiv.) and Et₃N (0.429 mL, 3.15 mmol, 2 equiv.) in anhydrous $CH_2Cl_2$ (3 mL) cooled at 0 °C was added dropwise. The dark red mixture was allowed to warm to room temperature slowly then stirred for a further 2 h. Once all starting material was consumed, the mixture was quenched with $H_2O$ (30 mL) and extracted with $CH_2Cl_2$ (2 × 30 mL). The combined organic layer was washed with 1 M NaHSO₄ aq. solution (30 mL), then concentrated to a red foam. The foam was then dissolved in $CH_2Cl_2$ (4 mL) and trifluoroacetic acid (2 mL) and stirred at room temperature for 1 h. Once the reaction was completed, the dark solution was evaporated. The resulting residue was extracted between $CH_2Cl_2$ (30 mL) and saturated NaHCO₃ aq. solution (30 mL), and the aqueous layer was back-extracted with $CH_2Cl_2$ (30 mL). The combined organic layer was concentrated to a dark yellow foam. The foam was suspended in $CH_2Cl_2$ (2 mL) and cyclohexane (4 mL) was slowly added to the mixture with constant stirring. After stirring for 10 min, the resulting yellow precipitate was collected via vacuum filtration and dried to afford the title compound as a yellow solid (556.1 mg, 71% yield). ¹H NMR (400 MHz, DMSO-*d₆*) δ 8.13 (d, *J* = 2.0 Hz, 1H), 7.87 − 7.80 (m, 1H), 7.69 (dd, *J* = 8.8, 2.0 Hz, 1H), 7.62 − 7.54 (m, 2H), 7.47 − 7.37 (m, 3H), 7.39 − 7.30 (m, 1H), 7.09 (s, 2H), 5.27 (s, 2H), 4.49 (d, *J* = 17.8 Hz, 1H), 4.33 (d, *J* = 17.8 Hz, 1H), 3.56 (s, 3H). ¹⁹F NMR (377 MHz, DMSO-*d₆*) δ −122.0. HRMS (ESI-TOF) *m/z*: calcd for $C_{20}H_{18}BrFN_2O_5SNa$ $[M^{79}Br + Na]^+$ 518.9996, found 519.0001; $[M^{81}Br + Na]^+$ 520.9977, found 520.9982.

## 5-(3-(Benzyloxy)−7-bromo-1-fluoronaphthalen-2-yl)−1,2,5-thiadiazolidin-3-one 1,1-dioxide (VIII).

To a solution of **VII** (528.4 mg, 1.06 mmol, 1 equiv.) in 12 mL anhydrous THF was added potassium *tert*-butoxide (150.2 mg, 1.34 mmol, 1.26 equiv.) and the resulting solution was stirred at room temperature for 1 h. Once complete, the reaction was quenched with 1 M HCl aq. solution (4 mL) and the mixture was extracted with EtOAc (20 mL). The aqueous layer was back-extracted with EtOAc (2 × 10 mL). The combined orange organic layer was washed with brine (30 mL), dried over MgSO₄, filtered and concentrated to a brown foam. *Tert*-butyl methyl ether (5 mL) was added to the solid and the resulting suspension was filtered under vacuum. The collected solid was further washed with *tert*-butyl methyl ether (5 mL) and dried to afford the title product as an off-white solid (447.2 mg, 90% yield). ¹H NMR (400 MHz, DMSO-*d₆*) δ 8.17 (d, *J* = 2.0 Hz, 1H), 7.91 − 7.85 (m, 1H), 7.74 (dd, *J* = 8.8, 2.0 Hz, 1H), 7.56 − 7.49 (m, 3H), 7.45 − 7.31 (m, 3H), 5.28 (s, 2H), 4.53 (s, 2H). ¹⁹F NMR (377 MHz, DMSO-*d₆*) δ −125.5. HRMS (ESI-TOF) *m/z*: calcd for $C_{19}H_{14}BrFN_2O_4SNa$ $[M^{81}Br + Na]^+$ 488.9715, found 488.9735.

## 5-(3-(Benzyloxy)−1-fluoro-7-(3-hydroxy-3-methylbutoxy)naphthalen-2-yl)−1,2,5-thiadiazolidin-3-one 1,1-dioxide (X).

A mixture of compound **IX** (447.2 mg, 0.961 mmol, 1 equiv.), RockPhos Pd G3 (32.2 mg, 0.0384 mmol, 0.04 equiv.) and Cs₂CO₃ (939.5 mg, 2.88 mmol, 3 equiv.) in a round bottom flask were subject to five vacuum and N₂ backfill cycles before being placed under N₂ atmosphere. A degassed preformed mixture of DMF (10 mL) and H₂O (52.9 μL, 2.88 mmol, 3 equiv.) was added to the mixture, and the flask was further subjected to 5 cycles of vacuum and N₂ backfill. The brown

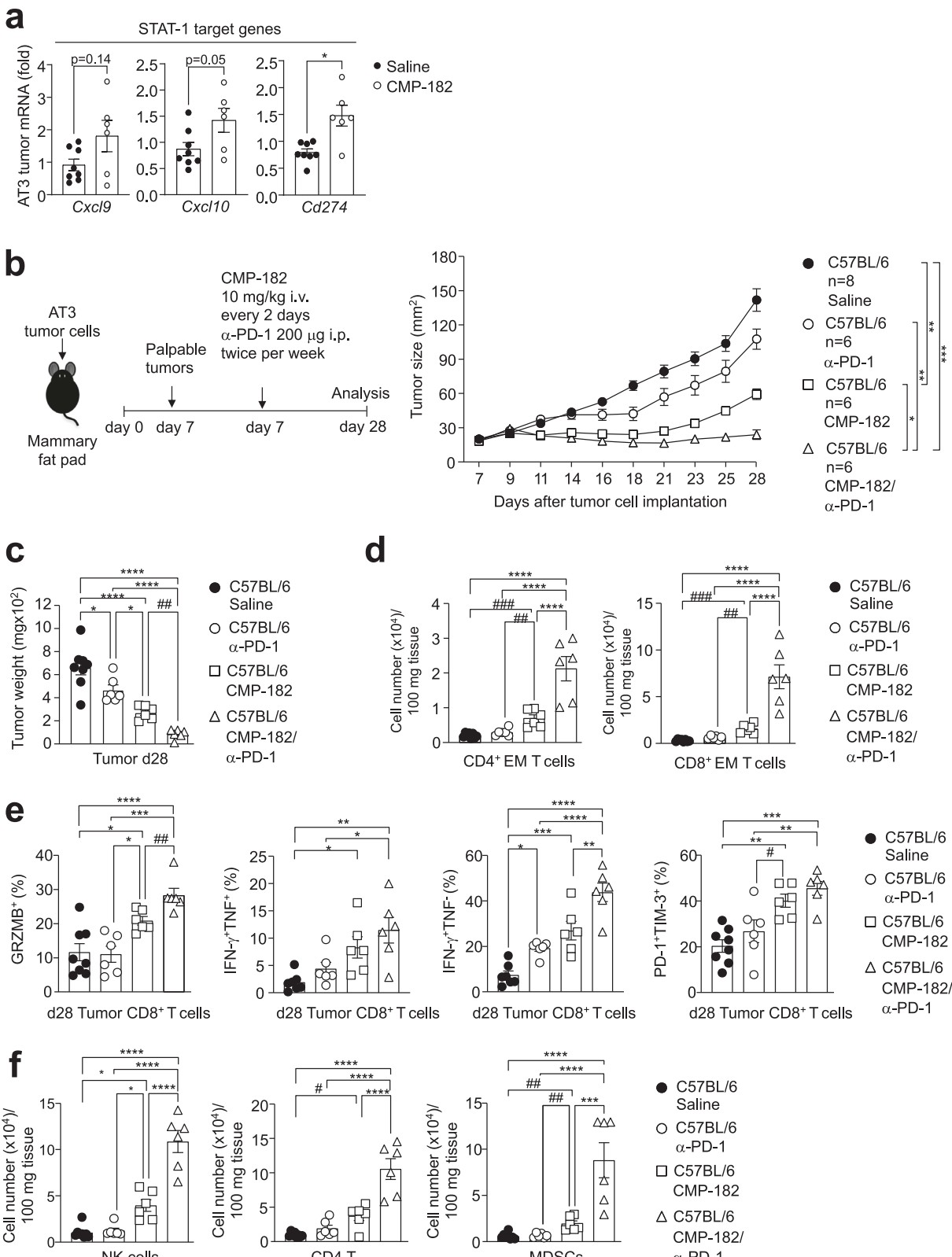

mixture was stirred at 80 °C for 16 h under a nitrogen atmosphere. Once the reaction was complete (monitored by LCMS, (ESI) $m/z$ 401.1 [M-H]⁻), the mixture was cooled to room temperature. 4-Bromo-2-methylbutan-2-ol (152.4 μL, 1.44 mmol, 1.5 equiv.) was added to the reaction mixture and stirred at room temperature for 16 h. Once complete, the mixture was extracted between EtOAc (30 mL) and 1 M HCl aq. solution (30 mL). The aqueous layer was back-extracted with

EtOAc (3 × 30 mL). The brown combined organic layer was washed with brine (30 mL), dried over Na₂SO₄, filtered, and concentrated to the crude product as a brown residue. The crude residue was then purified by automated C18 reverse phase column chromatography (40 g C18 silica gel, 25 μm spherical particles, eluent: 5% MeCN/H₂O (5 CV), gradient 5→100% MeCN/H₂O (20 CV), $t_R$ = 12.9 CV) to provide the product as a brown solid (252.2 mg, 53% yield). ¹H NMR (400 MHz,

**Fig. 9 | Compound 182 sensitizes AT3 tumors to PD-1 checkpoint blockade.**
**a** AT3 mammary tumor cells were injected into the fourth inguinal mammary fat pads of 8-week-old C57BL/6 female mice. Mice were treated with Compound 182 (CMP-182; 10 mg/kg i.v.; *n* = 6) or saline (*n* = 8) on days 10, 12, 14, 16, 18, 20, 22, 24, and 26 after tumor cell implantation; tumor growth curves are shown in Fig. 4d. The resulting AT3 tumors were processed for qPCR monitoring for the expression of STAT-1 target genes. **b**–**f** AT3 mammary tumor cells were injected into the fourth inguinal mammary fat pads of 8-week-old C57BL/6 female mice (*n* = 6–8 per group). Mice were treated with CMP-182 (10 mg/kg i.v.; *n* = 6) or saline (*n* = 8) on days 7, 9, 11, 13, 15, 17, 19, 21, 23, 25, and 27 and α-PD-1 (*n* = 6) twice per week (200 µg i.p.) after tumor cell implantation. **b** Tumor growth was monitored and **c** tumor weights were measured. **d**–**f** Tumor-infiltrating lymphocytes including

**d**, **f** CD44^hiCD62L^lo CD8^+ and CD4^+ effector/memory (EM) T cells, NK1.1^+TCRβ^− (NK) cells, CD4^+CD25^+FoxP3^+ regulatory T cells (T_regs) and granulocytic and monocytic CD11b^+F4/80^hi/loLy6C^+Ly6G^+/− myeloid-derived suppressor cells (MDSCs) were analyzed by flow cytometry. In (**e**) tumor-infiltrating T cells from (**b**) were stimulated with PMA/Ionomycin in the presence of Golgi Stop/Plug and stained for intracellular IFN-γ and TNF. Intracellular granzyme B (GRZMB), PD-1, and TIM-3 were detected in unstimulated tumor-infiltrating CD8^+ T cells. In (**a**–**f**) representative results (means ± SEM) from at least two independent experiments are shown. Significance for tumor sizes in (**b**) was determined using a 2-way ANOVA Test and for tumor weights in (**c**) using a 1-way ANOVA Test. In (**d**–**f**) significances were determined using a 1-way ANOVA Test and a 2-tailed Mann–Whitney U Test ($^\#p < 0.05$, $^{\#\#}p < 0.01$, $^{\#\#\#}p < 0.001$) where indicated.

DMSO-$d_6$) δ 7.78 (d, *J* = 9.0 Hz, 1H), 7.56 – 7.50 (m, 2H), 7.41 – 7.35 (m, 3H), 7.35 – 7.31 (m, 1H), 7.28 (d, *J* = 2.5 Hz, 1H), 7.25 – 7.19 (m, 1H), 5.23 (s, 2H), 4.21 (t, *J* = 7.1 Hz, 2H), 1.91 (t, *J* = 7.2 Hz, 2H), 1.19 (s, 6H). $^{19}$F NMR (377 MHz, DMSO-$d_6$) δ −126.9. HRMS (ESI-TOF) *m/z*: calcd for $C_{24}H_{25}FN_2O_6SNa$ [M+Na]$^+$ 511.1310, found 511.1326.

### 5-(1-Fluoro-3-hydroxy-7-(3-hydroxy-3-methylbutoxy)naphthalen-2-yl)−1,2,5-thiadiazolidin-3-one 1,1-dioxide (XI, Compound 182).

Compound **X** (250.0 mg, 0.511 mmol) and 10% Pd/C (54.5 mg, 0.0256 mmol, 0.05 equiv.) in a three-necked flask were subject to 5 cycles of vacuum and nitrogen backfill. Degassed MeOH (10 mL) was added to the flask and the mixture was stirred under H₂ atmosphere (balloon) at room temperature for 4 h. Once complete, the mixture was filtered through filter aid and the filter bed was washed with MeOH (20 mL). The brown filtrate was concentrated to a brown residue. The crude residue was then purified by automated C18 reverse phase column chromatography (40 g, C18 silica gel, 20-35 µm spherical particles, 100 Å, eluent: 5% MeCN/H₂O + 0.1% TFA (5 CV), gradient 5 → 100% MeCN/H₂O + 0.1% TFA (20 CV), $t_R$ = 9.8 CV) to provide the product as a light-brown solid (131.2 mg, 64% yield) after lyophilisation. $^1$H NMR (400 MHz, DMSO-$d_6$) δ 7.71 (d, *J* = 9.1 Hz, 1H), 7.22 (d, *J* = 2.5 Hz, 1H), 7.17 (dd, *J* = 9.0, 2.5 Hz, 1H), 7.07 (s, 1H), 4.54 (s, 2H), 4.19 (t, *J* = 7.2 Hz, 2H), 1.89 (t, *J* = 7.2 Hz, 2H), 1.19 (s, 6H). $^{13}$C{$^1$H} NMR (101 MHz, DMSO-$d_6$) δ 170.0, 155.8 (d, *J* = 253.7 Hz), 155.2, 151.2 (d, *J* = 2.6 Hz), 129.2 (d, *J* = 7.0 Hz), 127.8 (d, *J* = 2.9 Hz), 121.3, 118.0 (d, *J* = 14.8 Hz), 112.5 (d, *J* = 12.2 Hz), 106.3 (d, *J* = 3.4 Hz), 99.0 (d, *J* = 4.8 Hz), 68.0, 64.8, 55.4, 42.0, 29.7. HRMS (ESI-TOF) *m/z*: calcd for $C_{17}H_{19}FN_2O_6S$ [M+Na]$^+$ 421.0840, found 421.0839. HPLC: $t_R$ 4.710 min, 95% purity (254 nm).

### Cloning, expression, and purification of PTP proteins for enzymatic assays

cDNAs encoding recombinant PTPs (PTPN2, residues 1-387; PTP1B, residues 1-321; SHP-1, residues 245-543; SHP-2, residues 224-528; LYP/PTPN22, residues 1-294; STEP, residues 258-539; HePTP, residues 22-360; PTP-PEST, residues 5-304; FAP1, residues 2124-2485; PTPα, residues 173-793; PTPε, residues 107-697; CD45, residues 620-1236; CDC14A, residues 1-413; CDC14B, residues 1-411; MKP5, residues 320-647; VHZ, residues 1-150; Laforin, residues 1-331; LMWPTP, residues 1-158) were amplified by PCR and subcloned into the pET28a(+) bacterial expression vector to allow for the expression of N-terminal His-tagged proteins. E. coli BL21(DE3) (Novagen) was used as an expression host and the induction of protein expression was carried out in LB media with 1 mM IPTG at 18 °C overnight. Cell pellets were stored at −80 °C for subsequent protein purification.

Protein purification of PTPs for enzymatic assays was conducted at 4 °C. Frozen cell pellets were lysed by sonication in 40 ml cold lysis buffer (50 mM Tris-HCl, pH 8.0, 150 mM NaCl, 5 mM imidazole, and 1 mM PMSF) per litre cell pellet. Cell lysates were clarified by centrifugation using a Beckman JA-18 rotor for 15 min at 5308 x g. The supernatant was incubated with HisPur Ni-NTA resin (Thermo Scientific) for 2 h and then packed onto a column and washed with 50 resin volume of buffer A (50 mM Tris-HCl, pH 8.0, 500 mM NaCl, 5 mM imidazole). The His-tagged proteins were eluted with Buffer B (50 mM Tris-HCl, pH=8.0, 500 mM NaCl, 300 mM imidazole,). Pooled His-protein-containing fractions were concentrated, loaded onto a HiLoad 26/600 Superdex 75 column (GE Healthcare Biosciences) and eluted with storage buffer (50 mM Tris-HCl, pH 8.0, 150 mM NaCl, 1 mM DTT, 10% glycerol). Proteins used for inhibition assays were purified using Ni-NTA resin (Qiagen) followed by size exclusion column chromatography (ÄKTA pure, Cytiva) and the purity was determined to be >95% by SDS-PAGE and Coomassie staining. The protein was aliquoted and stored at −80 °C.

### Determination of IC$_{50}$, K$_i$, and inhibitor selectivity

The PTP kinetic assay was modified from previous reported protocols[73,74]. PTP activity was assayed using 6,8-Difluoro-4-Methylumbelliferyl Phosphate (DiFMUP, Invitrogen, cat# D6567,) as a substrate in 3,3-Dimethylglutaric acid (DMG) buffer (50 mM DMG, pH 7.0, 1 mM EDTA, 18 mM NaCl, 0.01% Triton X-100) at 25 °C. To determine the IC$_{50}$ values, the assays were performed in 96-well plates (Corning Costar 3915). The reaction was initiated by the addition of enzyme (for PTP1B and PTPN2 the final concentration was 0.05 nM, for other PTPs, the final concentration was 2.5 nM) to a reaction mixture containing DiFMUP (final concentration close to K$_m$ value; for PTP1B and PTPN2 this was 10 µM) and various concentrations of inhibitors. The final reaction volume is 200 µl. The reaction was allowed to proceed for 10 min and then quenched by the addition of 40 µl of a 160 µM solution of bpV(Phen) (Sigma-Aldrich, cat# SML0889). The fluorescence signal was measured using a CLARIOstar Plus Microplate Spectrophotometer (BMG Labtech) using excitation and emission wavelengths of 340 nm and 450 nm, respectively. Data were fitted using Prism GraphPad 9.2.0.

To determine the mode of inhibition, the reactions were initiated by the addition of enzyme (final concentration for PTP1B and PTPN2 was 0.05 nM) to the reaction mixtures (0.2 ml) containing various concentrations of DiFMUP with different concentrations of the inhibitor. The reaction was allowed to proceed for 10 min and then quenched by the addition of 40 µl of a 160 µM solution of bpV(Phen) (Sigma-Aldrich, cat# SML0889). The fluorescence signal was measured using a CLARIOstar Plus Microplate Spectrophotometer (BMG Labtech) using excitation and emission wavelength of 340 nm and 450 nm, respectively. Data were fitted using Prism GraphPad 9.2.0. For selectivity studies, the inhibition assay for other PTPs were performed under the same conditions as used for PTP1B and PTPN2 except with different DiFMUP concentration corresponding to the $K_m$ of the PTP studied[75].

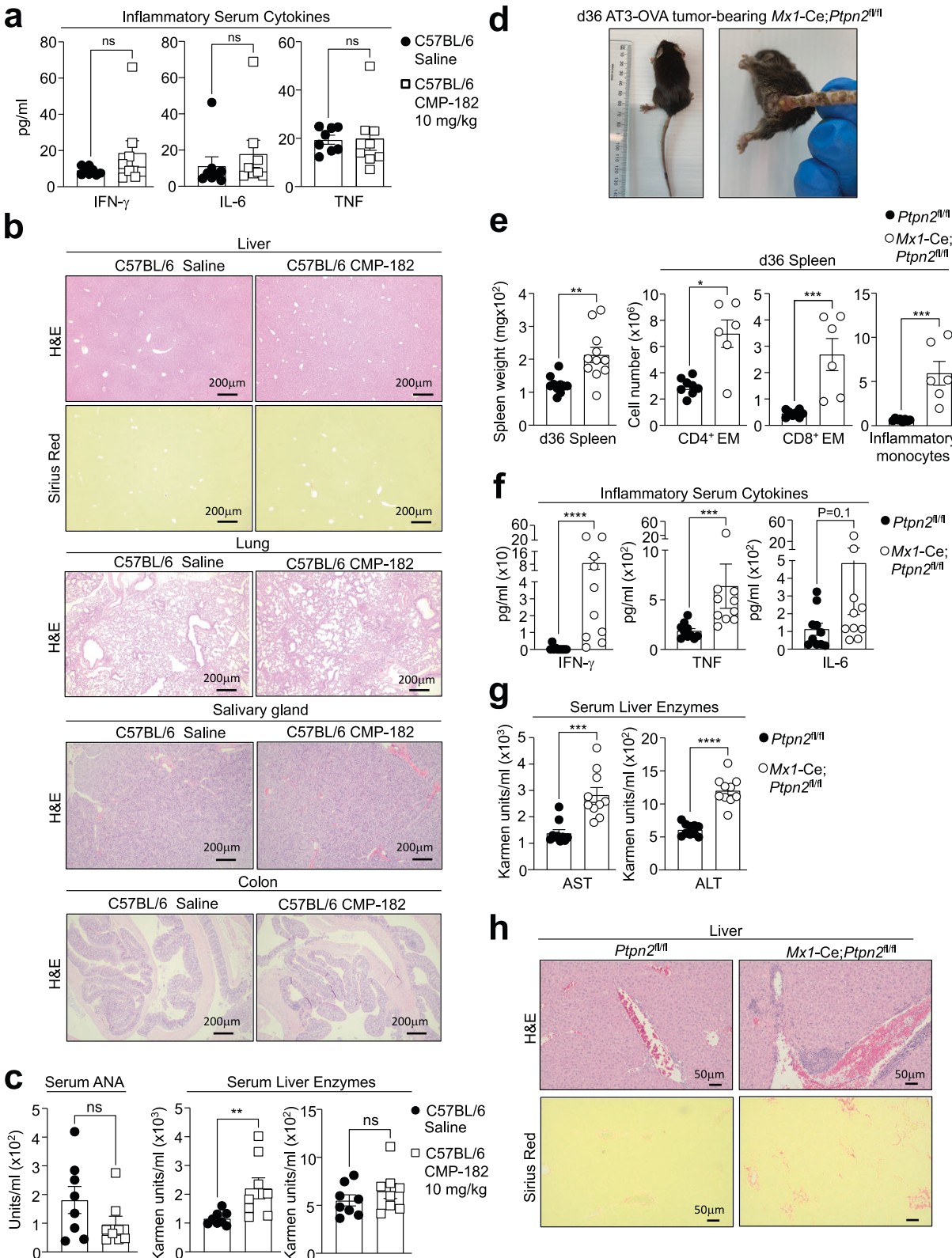

### Jump-dilution assays

In 3,3-Dimethylglutaric acid (DMG) buffer (50 mM DMG, pH 7.0, 1 mM EDTA, 18 mM NaCl, 0.01% Triton X-100), PTP1B (2.5 nM) or PTPN2 (2.5 nM) were preincubated with Compound 182 at different concentrations for 30 min followed by 50-fold dilution into DiFMUP (100 μM, 10-fold of $K_m$ for PTP1B and PTPN2) in DMG buffer. The reaction was allowed to proceed for 10 min under room temperature and was quenched by addition of a 160 μM solution of bpV(Phen). The fluorescence signal was measured using a CLARIOstar Plus Microplate Spectrophotometer (BMG Labtech) using excitation and emission wavelengths of 340 nm and 450 nm, respectively. Data were fitted using Prism GraphPad 9.2.0.

**Fig. 10 | Compound 182 does not promote systemic inflammation and auto-immunity. a**–**c** AT3-OVA mammary tumor cells were implanted into the fourth inguinal mammary fat pads of 8-week-old C57BL/6 female mice. Mice were treated with Compound 182 (CMP-182; 10 mg/kg i.v.; $n = 8$) or saline ($n = 8$) on days 6, 8, 10, 12, 14, 16, 18, and 21 after tumor cell implantation. **a** Serum cytokines were determined by flow cytometry using a BD Cytokine Bead Array (BD Biosciences. **b** Livers, lungs, salivary glands, and colons were fixed in formalin and processed for histological assessment (hematoxylin and eosin: H&E). Livers were also processed for Sirius Red staining. **c** Serum anti-nuclear antibodies (ANA) and serum liver enzymes AST and ALT in CMP-182-treated mice. **d**–**h** AT3-OVA mammary tumor cells were implanted into the fourth inguinal mammary fat pads of *Mx1*-Cre;*Ptpn2*^fl/fl mice. Mice were treated with poly I:C (250 μg/kg i.v.) to inducible delete PTPN2 on day 7, 9, and 11 after tumor cell implantation; the resultant tumor growth curves are shown in Fig. 6a. **d** Gross phenotype of *Mx1*-Cre;*Ptpn2*^fl/fl mice. **e** Spleen weight ($n = 9$-11 per group) and splenic CD44^hiCD62L^lo CD8^+ and CD4^+ effector/memory (EM) T cells and inflammatory monocytes in *Mx1*-Cre;*Ptpn2*^fl/fl ($n = 6$-8 per group) mice. **f** Serum cytokines in *Mx1*-Cre;*Ptpn2*^fl/fl mice ($n = 10$ per group) were determined by flow cytometry using a BD Cytokine Bead Array (BD Biosciences). **g** Serum liver enzymes AST and ALT in *Mx1*-Cre;*Ptpn2*^fl/fl mice ($n = 10$ per group). **h** Livers from *Mx1*-Cre;*Ptpn2*^fl/fl mice were fixed in formalin and processed for histological assessment (hematoxylin and eosin: H&E; Sirius Red). In (**a**, **c**, **e**–**g**) representative results (means ± SEM) from at least two independent experiments are shown. In (**b** and **h**) micrographs are representative of tissues from at least 5 mice per group. Significances were determined using a 2-tailed Mann–Whitney U Test.

## Label-free thermal shift assays (nanoDSF)
Label-free thermal shift assay experiments were performed with a Tycho NT.6 instrument (NanoTemper Technologies) based on a reported protocol[76]. The samples that contain 5 μM PTPN2/PTP1B and 50 μM Compound 182 were prepared with (DMG) buffer. Samples were heated in a glass capillary at a rate of 30 K/min and internal fluorescence at 330 and 350 nm was recorded. All samples were measured in triplicate. Data analysis, data smoothing, and the calculation of derivatives were done using the internal evaluation features of the Tycho instrument.

## Crystallization and structure determination of PTP1B in complex with Compound 182
**Protein expression and purification.** BL21(DE3) cells transformed with pGEX-4T encoding GST-(TEV)-PTP1B (2-321) were grown in S-broth plus 100 ug/ml Ampicillin at 37 °C, 180 rpm. At OD600 of 1, expression was induced with 1 mM IPTG at 18 °C, and growth continued overnight. Cells were harvested by centrifugation and resuspended in 20 mM Tris pH7.5, 300 mM NaCl, 2 mM TCEP, 1 mM PMSF, 5 mM Phenyl phosphate, DNase I and ~20 mg Lysosome. Cells were lysed by sonication, clarified by centrifugation, and the supernatant was applied to Glutathione Agarose (UBPBio). Beads were washed with 20 column volumes of the resuspension buffer, and PTP1B was eluted by TEV digest at 4°C overnight in 20 mM Tris, 300 mM NaCl, 2 mM TCEP. The eluate was concentrated and applied to Superdex200 size exclusion column (Cytiva) equilibrated in 20 mM Tris pH7.5, 300 mM NaCl, 2 mM TCEP. The Purest fractions as analyzed by SDS-PAGE were pooled and desalted into 20 mM Tris-HCl, pH 8.5, 50 mM NaCl before being applied to a MonoQ (HR5/5 Cytivia) equilibrated in 20 mM Tris-HCl, pH 8.5, 50 mM NaCl. PTP1B was eluted with a gradient from 20 mM Tris-HCl pH 8.5, 50 mM NaCl to 20 mM Tris-HCl pH 8.5, 700 mM NaCl over 20 column volumes. The cleanest fractions were pooled and concentrated to A280 12.6, TCEP was added to 2 mM and the protein was snap frozen in liquid nitrogen and stored at −80 °C.

**Crystallization and structure determination.** Crystallization was accomplished by hanging-drop vapor diffusion at 4 °C, using protein at 7-10 mg/ml, and a drop ratio of 1:1 protein: precipitant. Protein was mixed with 2-fold molar excess of 182 before setting down the trays. Crystals grew in 0.2 M Ammonium Tartrate, 20 %w/v PEG 3350 (Precipitant). They were cryo-protected in mother liquor with an additional 25% (v/v) ethylene glycol and flash frozen in liquid nitrogen. Diffraction data were collected on beamline MX2 at the Australian Synchrotron[77] using a wavelength of 0.9537 Å. Data were integrated using XDS[78]. The data used for refinement were cut at 1.55 Å. The data in the highest resolution shell had an $I/\sigma$ of 1.91 and a $CC\frac{1}{2}$ of 0.847 (Table 2). A molecular replacement solution was obtained with PHASER[79] as executed in PHENIX, using PDB 7MKZ[80] as the search model. Refinement was performed using PHENIX[81] and model building was performed in COOT[82]. Refinement converged to $R_{work}$ and $R_{free}$ values of 0.1747 and 0.1955, respectively. The final refined model had 97.64% residues in the favored region and 2.02% residues in the allowed region of the Ramachandran plot.

## Materials
CD3ε (BD Biosciences Cat# 553058) and CD28 (BD Biosciences Cat# 557393) antibodies for T cell stimulation and NK1.1 (BD Biosciences Cat# 564143) antibody for NK cell stimulation were purchased from BD Biosciences. Ly49H (clone 3D10, BioLegend Cat# 144702) antibody for NK stimulation was from BioLegend. For analyzing T cell signaling, p-(Y418) SFK (Thermo Fisher Scientific Cat# 14-9034-82) was from Invitrogen and p-(Y694) STAT-5 (D47E7) XP® (Cell Signaling Technology Cat# 4322) was from Cell Signaling. Recombinant human IL-2, recombinant murine IL-7 and recombinant human IL-15 were purchased from the PeproTech, recombinant mouse IL-12 and recombinant human IL-18 were from Australia Biosearch and SIINFEKL (N4) from Mimotopes. *InVivo*MAb anti-mouse PD-1 (Bio X Cell Cat# BE0146) and *InVivo*MAb rat IgG2a isotype control (Bio X Cell Cat# BE0251) were purchased from Bio X Cell. DNase I was from Sigma-Aldrich. Rabbit monoclonal PTP1B (Abcam Cat# ab244207) antibody for flow cytometry was from Abcam and mouse monoclonal PTPN2 6F3 antibody for immunoblotting and flow cytometry was from MédiMabs (Cat# MM-019-P). Mouse anti-actin (Thermo Fisher Scientific Cat# MA5-11866) from Invitrogen and mouse α-tubulin (Sigma-Aldrich Cat# T5168) was from Merck. The mouse anti-nuclear antibodies Ig's (total IgA+G+M) ELISA Kit was from Alpha Diagnostic Int. and the Transaminase II Kit from Wako Pure Chemicals. FBS was purchased from Thermo Scientific, RPMI 1640, DMEM, MEM, HEPES, non-essential amino acids, penicillin/streptomycin, and sodium-pyruvate were from Invitrogen, collagenase type IV was purchased from Worthington Biochemical and ABBV-CLS-484 was from MedChemExpress (Cat# HY-145923).

## Mice
Mice were maintained on a 12 h light-dark cycle in a temperature-controlled high-barrier facility with free access to food and water. Aged- and sex-matched littermates were used in all experiments; unless otherwise indicated, female donor mice were used for all in vivo experiments, whereas mice of the same sex were used for ex vivo experiments. *Ptp1b*^fl/fl, *Ptpn2*^fl/fl, *Lck*-Cre;*Ptp1b*^fl/fl, *Lck*-Cre;*Ptpn2*^fl/fl and *Mx1*-Cre;*Rosa26*-eYFP;*Ptpn2*^fl/fl (referred to as *Mx1*-Cre;*Ptpn2*^fl/fl) mice on a C57BL/6 background have been described previously[21,25]. *Ncr1*-Cre (C57BL/6) mice[56] were crossed with *Ptp1b*^fl/fl (C57BL/6) or *Ptpn2*^fl/fl (C57BL/6) to generate *Ncr1*-Cre;*Ptp1b*^fl/fl and *Ncr1*-Cre;*Ptpn2*^fl/fl mice respectively. C57BL/6/J mice were purchased from Monash Animal Research Platform (MARP, Clayton, Australia), the WEHI Animal Facility (Kew, Australia) or from the Animal Resource Centre (ARC, Perth, Australia).

## Tumor cell lines
The C57BL/6 mouse mammary tumor cell line AT3 and the corresponding cells engineered to express chicken ovalbumin (AT3-OVA) have been described previously[11]. MC38 colon adenocarcinoma cells[11]

and the corresponding $B2m^{-/-}$ MC38 cells[57] have also been described previously. PTPN2 was deleted in AT3 or AT3-OVA mammary tumor cells using CRISPR Ribonucleoprotein (RNP) gene-editing as described previously[10]. P3 Primary Cell 4D-Nucleofector X Kits (Cat#. V4XP-3032, Lonza), including 4D-Nucleofector Solution, 16-well Nucleocuvette™ Strips, were prepared and used as per manufacturer's instructions. Briefly, short guide (sg) RNAs (600 pmol; Synthego) targeting the *Ptpn2* locus (*Ptpn2*; 5'-AAGAAGUUACAUCUUAACAC) or non-targeting sgRNAs (GCACUACCAG AGCUAACUCA) were precomplexed with recombinant Cas9 (74 pmol; Alt-R S.p. Cas9 Nuclease V3, IDT). Cells $(4.5 \times 10^5)$ were resuspended in 4D-Nucleofector Solution and then mixed with the Cas9/sgRNA complex and electroporated using the 4D-Nucleofector X Unit (Lonza). All tumor cell lines were maintained in high-glucose DMEM supplemented with 10% (v/v) FBS, L-glutamine (2 mM), penicillin (100 units/mL)/streptomycin (100 µg/mL) and HEPES (10 mM). All tumor cell lines were authenticated for their antigen/marker expression by flow cytometry. Cells were routinely tested for *Mycoplasma* contamination by PCR and maintained in culture for less than four weeks.

## Immunoblotting

Tumor cells were washed with ice-cold PBS and lysed in ice-cold RIPA lysis buffer [50 mM HEPES pH 7.4, 1% (v/v) Triton X-100, 1% (v/v) sodium deoxycholate, 0.1% (v/v) SDS, 150 mM NaCl, 10% (v/v) glycerol, 1.5 mM $MgCl_2$, 1 mM EGTA, 50 mM sodium fluoride, 1 µg/ml pepstatin A, 5 µg/ml aprotinin, 1 mM benzamadine, 1 mM phenylmethysulfonyl fluoride, 1 mM sodium vanadate], clarified by centrifugation (16,000 x g, 10 min, 4°C), resolved by SDS-PAGE (10%), transferred to PVDF (Merck, Darmstadt. Germany) and immunoblotted as indicated.

## Flow cytometry

Single-cell suspensions from spleens and lymph nodes were obtained as described previously[11]. For the detection of intracellular cytokines and granzyme B, T cells were fixed and permeabilized with the BD Cytofix/Cytoperm kit according to the manufacturer's instructions. Briefly, T cells were first surface stained with fluorophore-conjugated antibodies and then fixed followed by permeabilization on ice for 20 min. Fixed and permeabilized cells were stained for IFN-γ, TNF, and granzyme B for 45 min at room temperature. For the detection of intracellular FoxP3 the Foxp3/Transcription Factor Staining Buffer Set (eBioscience) was used according to the manufacturer's instructions. Briefly, cells were fixed and permeabilized for 1 h on ice followed by intracellular staining at room temperature for 45 min. For the detection of serum cytokines, the BD CBA Mouse Inflammation Kit™ was used according to the manufacturers' instructions.

For detecting intracellular PTP1B splenic $CD3^+CD19^-CD49b^-NK.1.1^-$ T cells and $CD3^-CD19^-CD49b^{hi}NK.1.1^{hi}$ NK cells were blocked with anti-CD16/CD32 (BD Biosciences, Cat# 553142) and then fixed and permeabilized with the BD Cytofix/Cytoperm kit according to the manufacturer's instructions. Briefly, surface-stained and permeabilised cells were stained intracellularly with the PTP1B monoclonal rabbit antibody (1/200, 45 min, room temperature) which was detected by the secondary antibody against rabbit IgG (H + L) F(ab')$_2$ fragment conjugated to Alexa Fluor 647 (Jackson ImmunoResearch Labs) (30 min, room temperature). For detecting intracellular PTPN2 splenic $CD3^+CD19^-CD49b^-NK.1.1^-$ T cells and $CD3^-CD19^-CD49b^{hi}NK.1.1^{hi}$ NK cells were blocked with anti-CD16/CD32 and then fixed in 200 µl Cytofix™ Fixation Buffer (BD Biosciences) for 15 min at 37 °C. Cells were washed twice with PBS and permeabilized in 200 µl methanol/acetone (50:50) at −20 °C overnight and then stained with the 6F3 PTPN2 monoclonal mouse antibody (1/100, 30 min, room temperature) which was detected by the secondary antibody against a mouse IgG (H + L) F(ab')$_2$ fragment conjugated to AlexaFluor 647 (Molecular Probes, Thermo Fisher Scientific).

Cells $(1 \times 10^6/10 \text{ µl})$ were stained with the specified antibodies on ice for 30 min in D-PBS supplemented with 2% FBS in the dark and analyzed using a Fortessa or Symphony A3 (BD Biosciences). Data was acquired using BD FACS DIVA Software v8.0.1 and analyzed using FlowJo10 (Tree Star Inc.) software. Lymphocytes were gated for eYFP$^+$ cells to discriminate between PTPN2 sufficient and PTPN2 negative cells after poly(I:C)-induced PTPN2-deletion in *Mx1*-Cre;*Ptpn2*$^{fl/fl}$ mice. For cell quantification, known numbers of Flow-Count Fluorospheres (Beckman Coulter) were added to samples before analysis. Dead cells were excluded with propidium iodide (1 µg/ml; Sigma-Aldrich). Paraformaldehyde fixed dead cells were excluded with the LIVE/DEAD™ Fixable Near-IR Dead Cell Stain Kit (Thermo Fisher Scientific).

The following antibodies from BD Biosciences, BioLegend, Invitrogen or eBioscience were used for flow cytometry: Phycoerythrin (PE) or peridinin chlorophyll protein -cyanine 5.5 (PerCP-Cy5.5)-conjugated CD3 (BD Biosciences Cat#5 61808, BD Biosciences Cat# 55163); Pacific Blue-conjugated (PB), PerCP-Cy5.5 or phycoerythrin-cyanine 7 (PE-Cy7)-conjugated CD4 (BD Biosciences Cat# 558170; BD Biosciences Cat# 550954, BD Biosciences Cat# 561099); Brilliant Violet (BV) 711, Allophycocyanin (APC), PB or Brilliant Ultra Violet (BUV) 395-conjugated CD8 (BioLegend Cat# 100759; BD Biosciences Cat# 558106; BD Biosciences Cat#565968); BUV805 or BV711-conjugated CD11b (BD Biosciences Cat# 741934; BioLegend Cat# 101241); APC-conjugated CD11c (BioLegend Cat# 117309); PE-Cy5 or PE-Cy7-conjugated CD19 (BioLegend Cat# 115510, BD Biosciences Cat# 552854); PE-conjugated CD25 (BD Biosciences Cat# 553866); BV650-conjugated CD27 (BD Biosciences Cat# 740491); Fluorescein isothiocyanate (FITC) or BV786-conjugated CD44 (BD Biosciences Cat# 553133; BD Biosciences Cat#563736); APC or APC-eFluor780-conjugated CD45 (Thermo Fisher Scientific Cat# 17-0451-83, Thermo Fisher Scientific Cat# 47-0451-82); BUV563-conjugated CD49b (BD Biosciences Cat# 741280); BV480-conjugatd Ly-49H (BD Biosciences Cat# 746493); APC, BV421, PE or BUV737-conjugated CD62L (BD Biosciences Cat# 553152; BioLegend Cat# 104435; BD Biosciences Cat# 553151; BD Biosciences Cat# 612833); PE-conjugated CD107a (BD Bioscience Cat# 558661); FITC or PE-Cy7-conjugated CD279 (BioLegend Cat# 135213; BioLegend Cat# 135216); PE-Cy7-conjugated CD335 (BioLegend Cat# 137618); PE-Cy7-conjugated CD366 (BioLegend Cat# 134010); BD Biosciences Cat# 552094); BV605 or PerCP-Cy5.5-conjugated TCR-β (BioLegend Cat# 109241; BD Biosciences Cat# 560657); BV421-conjugated NK1.1 (PK136; BD Biosciences Cat# 562921); APC-Cy7-conjugated Ly6C (BD Biosciences Cat# 560596); FITC-conjugated Ly6G (BD Biosciences Cat# 561105); PE or PE-Cy5-conjugated F4/80 (BD Biosciences Cat# 565410, BioLegend Cat# 123112); PE-Cy7-conjugated IFNγ (BD Biosciences Cat# 557649, Thermo Fisher Scientific Cat# 25-7311-82); APC-conjugated TNF (BD Biosciences Cat# 561062); Alexa Fluor 647 or BV421-conjugated Granzyme B (BioLegend Cat# 515405, BD Biosciences Cat#563389); FITC-conjugated Foxp3 (Thermo Fisher Scientific Cat# 11577382); eFlor660-conjugated phospho-SRC (Tyr418) (Invitrogen Cat# 50-9034-42).

## T cell activation assay

For assessing T cell activation markers in vitro, single-cell suspensions of OT-I CD8$^+$ cells obtained from the lymph nodes of wild type or PTPN2-deficient (*Lck*-Cre;*Ptpn2*$^{fl/fl}$) mice were stimulated with 1 nM of the cognate ovalbumin (OVA) peptide antigen SIINFEKL (N4) in the presence of vehicle (1% v/v DMSO) or Compound 182 (1 µM) for 48 h in complete T cell media [RPMI-1640 supplemented with 10% (v/v) FBS, L-glutamine (2 mM), penicillin (100 units/ml)/streptomycin (100 µg/ml), MEM non-essential amino acids (0.1 mM), sodium-pyruvate (1 mM), HEPES (10 mM) and 2-β-mercaptoethanol (50 µM)]; vehicle or Compound 182 (1 µM) were added every 24 h. The cell surface markers CD8, CD25, and CD69 were analyzed using fluorophore-conjugated antibodies by flow cytometry.

For the detection of intracellular IFNγ, TNF and Granzyme B, single-cell suspensions were obtained from lymph nodes of wild type (OT-I;*Ptpn2*fl/fl) or PTPN2-deficient (OT-I;*Lck*-Cre;*Ptpn2*fl/fl) mice and stimulated with 1 nM N4 peptide for 16 h and rested in complete T cell media supplemented with IL-2 (5 ng/ml) and IL-7 (0.2 ng/ml) for 2 days. On day 3, OT-I;*Ptpn2*fl/fl versus OT-I;*Lck*-Cre;*Ptpn2*fl/fl T cells were treated with vehicle (1% v/v DMSO) or Compound 182 (1 μM) overnight and then stimulated with 1 nM N4 in the presence of vehicle or Compound 182 (1 μM) for 4 h at 37 °C in complete T cell media supplemented with BD GolgiPlug™ and BD GolgiStop™ (BD Biosciences). Cells were then stained with fluorophore-conjugated antibodies to cell surface CD8 and intracellular IFNγ, TNF, and Granzyme B and analyzed by flow cytometry.

### T cell proliferation

For analyzing cellular proliferation in vitro, single-cell suspensions from lymph nodes of wild type (OT-I;*Ptpn2*fl/fl) or PTPN2-deficient (OT-I;*Lck*-Cre;*Ptpn2*fl/fl) mice were stained with 2 μM CellTrace™ Violet (CTV; Molecular Probes, Thermo Fisher Scientific) in PBS supplemented with 0.1% (v/v) FBS for 20 min at 37 °C. Cells were then washed with ice-cold PBS supplemented with 10% (v/v) FBS three times and 200,000 OT-I cells were stimulated 0.5 nM SIINFEKL (N4) peptide in the presence of vehicle (1% v/v DMSO) or Compound 182 (1 μM) for 3 days in complete T cell medium at 37 °C. On day 3, OT-I cells were harvested and CTV dilution was monitored by flow cytometry. For cell number quantification, known numbers of Flow-Count Fluorospheres (Beckman Coulter) were added to samples before analysis.

### TCR and cytokine signaling

For assessing TCR signaling, single-cell suspensions of naïve (CD44$^{lo}$CD62L$^{hi}$) lymph node CD8$^+$ T cells (isolated using the EasySep™ Mouse Naïve CD8$^+$ T cell Isolation Kit, Stemcell Technologies) were washed with ice-cold PBS supplemented with 2% (v/v) FBS and stained with fixable viable dye and fluorophore-conjugated antibodies to CD8 and CD4 on ice for 15 min. Cells were then washed and pre-treated with vehicle (1% v/v DMSO) or Compound 182 (10 μM) in complete T cell medium supplemented with 1% (v/v) FBS at 37 °C for 4 h with agitation. Cells were then incubated with 5 μg/ml hamster α-mouse CD3ε for 20 min on ice, washed with complete T cell medium supplemented with 1% (v/v) FBS, and then TCR cross-linked with goat α-hamster IgG (20 μg /ml, Sigma) for the indicated times at 37 °C. The reaction was terminated by the addition of paraformaldehyde (BD Cytofix™ Fixation Buffer) to a final concentration of 2% (v/v) on ice for 20 min. Fixed cells were washed in complete T cell medium/1% (v/v) FBS and permeabilised with 90% (v/v) methanol on ice for 30 min. Permeabilised cells were stained with fluorophore-conjugated antibodies to Y418 phosphorylated and activated SFKs (p-SFK; 1:100, Invitrogen) for 1 h at room temperature; p-SFK mean fluorescence intensities in CD8$^+$ T cells were determined by flow cytometry.

For assessing IL-2-induced STAT-5 Y694 phosphorylation (p-STAT-5) in activated OT-I cells, single cell suspension of OT-I cells from the lymph nodes of OT-I;*Ptpn2*fl/fl mice were stimulated with 1 nM SIINFEKL (N4) for 16 h. Cells were then washed with RMPI-1640 twice and pre-treated with vehicle (1% v/v DMSO) or Compound 182 (1 μM) for 1 h and then stimulated with 5 ng/ml IL-2 for the indicated times at 37 °C in the presence of vehicle or Compound 182. Reactions were terminated by washing the cells with ice-cold PBS and then stained with fixable viable dye and fluorophore-conjugated antibodies to CD8 on ice for 15 min. Cells were then washed and fixed with 4% (w/w) paraformaldehyde (BD Cytofix™ Fixation Buffer) at room temperature for 30 min and permeabilised with 90% (v/v) methanol at −20 °C for 30 min. Permeabilised cells were incubated with a rabbit antibody specific for p-STAT-5 (p-STAT-5; D47E7, Cell Signaling Technology) for 60 min at room temperature, washed thrice with PBS and then stained with goat anti-rabbit IgG (H + L) F(ab′)$_2$ fragment conjugated to Alexa

Fluor 647 (Jackson ImmunoResearch Labs; West Grove, PA) to monitor for p-STAT-5 in CD8$^+$ OT-I cells by flow cytometry.

### Tumor studies

Male or female mice were injected subcutaneously with $5 \times 10^5$ MC38 tumor cells resuspended in 100 μl D-PBS into the flank. Alternatively, female mice were injected orthotopically with $5 \times 10^5$ AT3 or AT3-OVA or PTPN2-deleted AT3 or AT3-OVA tumor cells resuspended in 20 μl D-PBS into the fourth mammary fat pad. Tumor sizes were measured in two dimensions (mm$^2$) using electronic callipers two to three times per week as soon as tumors were palpable (20–25 mm$^2$). Mice were sacrificed at the ethical experimental endpoint (the maximal tumor size permitted was 200 mm$^2$ and this was not exceeded) and individual tumors were removed to measure tumor weights or to assess immune cell infiltrates by flow cytometry.

### Immunohistochemistry

For the immunohistochemistry analysis of mammary tumor samples, samples were dissected and fixed in 10% formalin. After fixation, the tissues were embedded in paraffin and sectioned onto microscopic glass slides at 4 μm thickness per section. After deparaffinization and rehydration, the slides were subjected to antigen retrieval in Tris/EDTA buffer (pH 8.0) at 120 °C for 10 min and then cooled down to room temperature. The slides were blocked with 5% (v/v) normal goat serum (NGS) containing 0.1% (v/v) TritonX-100 in PBS for 1 h at room temperature and then incubated with primary antibody p-STAT-1 (Tyr701) (1:500; Cell Signaling Technology), or CD3ε (1:1000; Dako) at room temperature for 1 h. Slides were then washed with PBS containing 0.05% (w/v) Tween-20 and visualized using a rabbit IgG VECTORSTAIN ABC Elite and DAB (3,3′-diaminobenzidine) Peroxidase Substrate Kit (Vector Laboratories, UK) and counterstained with haematoxylin. Slides were imaged on an Olympus CX43 microscope (Olympus, Tokyo, Japan).

### Compound 182 treatment and immune checkpoint blockade

Mice with established tumors (30–50 mm$^2$) were randomly grouped to receive 0.9% (w/v) saline only, or Compound 182 dissolved in 0.9% (w/v) saline. Compound 182 was administered at 10 mg/kg body weight in 200 μl every second day intravenously. Control animals received an injection at the same time of 200 μl of saline. Stock solutions were stored at −80 °C and thawed before each administration. Mice were sacrificed at the experimental endpoint (the maximal tumor size permitted was 200 mm$^2$ and this was not exceeded) and individual tumors were removed to measure tumor weights or to assess immune cell infiltrates by flow cytometry.

To assess the combined effects of PTPN2/PTP1B-inhibition with Compound 182 and immune checkpoint blockade with α-PD-1, tumor-bearing mice (30–50 mm$^2$) were treated with Compound 182 as described above every second day. In addition, mice received four intraperitoneal injections of α-PD-1 (200 μg in 200 μl D-PBS; clone RMP1-14), or rat IgG2a isotype control (100 μg in 200 μl D-PBS, Bio X Cell) twice per week. Mice were sacrificed at the experimental endpoint (the maximal tumor size permitted was 200 mm$^2$ and this was not exceeded) and individual tumors removed to measure tumor weights or to assess immune cell infiltrates by flow cytometry.

### Analysis of tumor-infiltrating T cells

Tumor-bearing mice were sacrificed and tumors were excised and digested at 37 °C for 30 min using a cocktail of 1 mg/ml collagenase type IV (Worthington Biochemicals) and 0.02 mg/ml DNase I (Sigma-Aldrich) in DMEM supplemented with 2% (v/v) FBS. Cells were passed through a 70 μm cell strainer (BD Biosciences) twice and processed for flow cytometry. For assessing T cell activation, cells were surface stained with fluorophore-conjugated antibodies to CD62L, CD44, PD-1, and TIM-3. For the assessment of NK cells, B cells, myeloid-derived

suppressor cells (MDSCs), dendritic cells (DCs), and tumor-associated macrophages, cells were surface stained with fluorophore-conjugated antibodies to NK1.1, TCRβ, CD19, CD11b, CD11c, F4/80, Ly6C and Ly6G. For the detection of intracellular cytokines TNF and IFNγ, T cells were stimulated with PMA (20 ng/ml; Sigma-Aldrich) and ionomycin (1 µg/ml; Sigma-Aldrich) in the presence of GolgiPlug™ and GolgiStop™ (BD Biosciences) for 4 h at 37 °C in complete T cell medium. For the intracellular detection of granzyme B, cells were processed untreated without the addition of BD GolgiPlug™ and BD GolgiStop™.

## Quantitative real-time PCR

For quantitative real-time PCR analysis, RNA from tumors was isolated using RNAzol (Sigma-Aldrich, St. Louis, MO), and RNA quality was assessed using a NanoDrop 2000 (Thermo Fisher Scientific, Waltham MA). mRNA was reverse-transcribed using the High-Capacity cDNA Reverse Transcription Kit (Applied Biosystems, Thermo Fisher Scientific, Waltham, MA) and processed for quantitative Real-Time PCR using the Fast SYBR Green Master Mix (Applied Biosystems, Thermo Fisher Scientific, Waltham MA) and PrimePCR™ SYBR Green Assays (Bio-Rad, Hercules, CA) for *Cxcl9* (qMmuCID0023784), *Cxcl10* (qMmuCED0049500), *Tapbp* (qMmuCEP0035035), *Tap1* (qMmuCEP0057739), (qMmuCIP0031437), *H2ab1* (qMmuCIP0029038), *Cd274* (qMmuCED0044192), *H2k1* (qMmuCEP0034842), *Tnf* (qMmuCED0004141) and *Ifng* (qMmuCID0006268). Relative gene expression was determined by normalization to the house-keeping gene *Rps18* (qMmuCED0045430) or *Gapdh* (qMmuCED0027497). Gene quantification was performed with the comparative Ct (ΔΔCt) method.

## Animal ethics

All animal experiments were performed in accordance with the NHMRC Australian Code of Practice for the Care and Use of Animals. All protocols were approved by the Monash University School of Biomedical Sciences Animal Ethics Committee (Ethics numbers: 23177, 36697).

## Statistical analyses

Statistical analyses were performed with Graphpad Prism software 9 using the non-parametric 2-tailed Mann-Whitney U Test, the parametric 2-tailed Student's *t* test, the one-way or 2-way ANOVA-test using Kruskal–Wallis, Turkey, Bonferroni or Sidak post-hoc comparison where indicated. $*p < 0.05$, $**p < 0.01$, $***p < 0.001$ and $****p < 0.0001$ were considered as significant.

## Reporting summary

Further information on research design is available in the Nature Portfolio Reporting Summary linked to this article.

# Data availability

Data that support the findings of this study are available on request from the corresponding author. Crystal structure atomic coordinates have been deposited in the Protein Data Bank under accession code 8SKL (PTP1B in complex with 182). The source data underlying Figs. 2–5, 6a–d, f, 7a–d, 9a–f, 10a, c, e–g and Supplementary Figs. 6–13 are provided as a Source Data file. Data can also be accessed via the Monash University Research Repository (Figshare) [https://doi.org/10.6084/m9.figshare.23659113]. Data availability: Atomic coordinates for the crystal structure have been deposited in the Protein Data Bank under accession code 8SKL Source data are provided with this paper.

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

## Acknowledgements

We thank Crystal Stivala for her technical support. This work was supported by the National Health and Medical Research Council (NHMRC) of Australia (2008572 to T.T.; 1184615 to N.D.H. and T.T.), the US Army Department of Defense (W81XWH-21-1-0126 BC200609 to T.T.), and the US National Institutes of Health (RO1CA069202 to Z.-Y.Z.). This research was undertaken in part using the MX2 beamline at the Australian Synchrotron, part of ANSTO, and made use of the Australian Cancer Research Foundation (ACRF) Eiger.

## Author contributions

T.T. conceived the study. E.T. and J.B.B. synthesized Compound 182. T.T., F.W. J.J.B., N.D.H., J.J.B., N.J.K., and Z.Y.Z. designed the experiments. S.L., X.D., and H.S. performed cell biology experiments. S.L., H.S., and F.W. performed animal experiments. H.C., J.J.B., and N.J.K. performed structural biology experiments. N.J.K. built and refined structural models. J.D. and Z.Q. performed biochemical inhibitor experiments. All authors analyzed the results and contributed to figure preparation and the reviewing and editing of the manuscript. T.T. and F.W. wrote the manuscript.

## Competing interests

F.W. and T.T. are inventors on pending patents related to this work filed by Monash University and the Peter MacCallum Cancer Centre. T.T. is on the scientific advisory board of DepYmed. Z.Y.Z. is a co-founder and serves on the scientific advisory board of Tyligand Bioscience. N.D.H. is a founder and shareholder in oNKo-Innate. N.D.H. serves on an advisory board for Bristol Myers Squibb. The remaining authors declare no competing interests.
