## [Peer Review File · Nature Communications]

A small molecule inhibitor of PTP1B and PTPN2 enhances T cell anti-tumor immunityReviewers' Comments:

Reviewer #1:

Remarks to the Author:

This manuscript reports the synthesis, characterization, and anti-tumor activity of a dual inhibitor of the protein tyrosine phosphatases PTP1B and TCPTP (also known as PTPN1 and PTPN2). PTPs play critical roles in numerous cellular signaling pathways and have been proposed as intriguing therapeutic targets for decades, but no PTP inhibitor has made it into the clinic to date. PTP1B and TCPTP are particularly intriguing therapeutic targets, as they function as intracellular checkpoints in immune cell signaling. In this manuscript, the authors study a dual PTP1B/TCPTP inhibitor termed Compound 182 based upon a compound currently in clinical trials as monotherapy and sensitization for α -PD-1 therapy (ABBV-CLS-484). The authors demonstrate that Compound 182 can repress the growth of syngenic tumors in mice and enhance the response to PD-1 checkpoint blockade without overt immune-related toxicities. These results are noteworthy and likely to be of significance in the field both because they provide a dual-specificity inhibitor with anti-cancer properties and also because they highlight a novel strategy for enhancing PD-1 checkpoint blockade therapy in otherwise resistant tumors. The conclusions of the paper are well supported by the data provided. The methodology appears to be sound and enabling details are provided. Major and minor concerns that should be addressed prior to publication are outlined below.

Major concern:

1. The largest concern about this work centers around its novelty. The authors present this work as an investigation of a compound directly related to a clinical candidate already in Phase 1 trials through Calico Life Sciences and AbbVie. This clinical candidate (ABBV-CLS-484) has been disclosed and discussed publicly (see https://aacrjournals.org/cancerres/article/82/12_Supplement/ND06/703170/Abstract-ND06-ABBV-CLS-484-An-active-site-PTPN2-N1). It is hard to imagine that the industrial developers of these compounds do not have very similar data to that presented in this manuscript, and it appears that at least some of these data were presented at the American Association for Cancer Research Annual Meeting in April of 2022 (based upon the abstract). It does not appear that the data have been published yet, however, somewhat mitigating this concern.

Minor concerns:

1. The authors are encouraged to use a consistent nomenclature for the enzymes they are targeting (i.e. either PTP1B/TCPTP or PTPN1/PTPN2).
2. Several editorial mistakes are noted – the manuscript could use a thorough review for these types of minor mistakes. See, for example, lines 70, 106, 190, 431, 1105, 1172, etc.
3. Similarly, the figure captions should be checked carefully. Most appear to be accurate, but the caption for Figure 1 is mislabeled (subfigures A, B, C are not as indicated).
4. The compounds containing bromine should have two isotopes of nearly equal abundance, it would be a good idea to include the masses of both of the most prevalent isotopes in the HRMS data (lines 305, 731, etc.).
5. Compound 182 should be clearly labeled in Scheme 1 – it would be helpful to include the structure of ABBV-CLS-484 here as well, or at least to reference Figure S1 early on (line 145 or so).
6. In the discussion, it could be helpful to be explicit that “In this study we focused on Compound 182 since the structure of AbbVie’s clinical candidates ... had not been revealed when the work began...”
7. The enzyme kinetic data seem to have been collected in an unusual way. The substrate DiFMUP is commonly used in assays of PTP activity and inhibition because it is a sensitive fluorogenic assay that can give continuous readouts of activity. It is not clear why the authors chose to carry out the assays in a discontinuous manner. The exact concentrations of DiFMUP should be included in the experimental section, as well (see line 883). The IC₅₀ curves should be provided, both for Compound 182 and for ABBV-CLS-484 (see line 549). For ABBV-CLS-484, the authors should indicate how they obtained this sample. In addition, the inhibition of Compound 182 under standard conditions as well as under the conditions of the jump dilution should be shown in Figure 1B.

8. The selectivity of Compound 182 for PTPN1/PTPN2 over the panel of PTPs shown in Table 1 is impressive. However, for a true clinical candidate, the authors would need to investigate the activity of Compound 182 against other common targets (kinase panels, GPCR panels, etc.).

9. It would be helpful to explain the rationale behind dosing Compound 182 every 3 days – is this driven by PK/PD data, toxicity, etc.?

10. The data indicating that Compound 182 does not promote inflammation or autoimmunity are quite solid (Fig 12), although the timeframe is fairly short. It would be interesting to investigate whether longer exposure to the compound results in a different outcome.

Reviewer #2:

Remarks to the Author:

The paper by Liang and colleagues describes the development of a dual PTP1b/PTPN2 inhibitor and characterises its effects on immune cells and as an anti-cancer reagent. Overall the paper does a very thorough job of characterising this compound and demonstrates that it; a) is highly selective for PTPN2/PTP1b over other related phosphatases, b) promotes T cell activation via effects on TCR and cytokine receptor signalling, c) enhances T cell responses to tumours in vivo, d) reduces tumour growth in several different mouse models via effects primarily dependent on T cells, e) sensitises otherwise resistant tumours to anti-PD1 therapy, f) does not induce systemic inflammation despite potent anti-tumour function.

Overall, this is a very thorough paper with very convincing data that adds to evidence previously published by the same group suggesting that PTPN2 and or PTP1b are intracellular checkpoints for effective T cell anti-tumour responses and are potential therapeutic targets. The major novelty lies in the use of dual inhibitors that target both phosphatases; much of the biological effects of the inhibitor are as expected given the known functions of the phosphatases in T cells and previous studies showing that targeting either phosphatase alone through genetic means enhances anti-tumour immunity. However, the authors have done an excellent and thorough job and I struggle to find any significant weaknesses in the current work.

Reviewer #3:

Remarks to the Author:

This paper reports in vitro and in cell studies of a small-molecule inhibitor specific to PTP1B and TCPTP (PTPN2) over all other PTP enzymes. The molecule is very similar to one currently in clinical trials for the same protein targets. A crystal structure of the compound is determined, which goes some way toward explaining the high specificity of this compound toward these two PTPs. A multitude of additional experiments in human cell lines and mouse models are employed to show that the inhibitor combats tumor growth by enhancing T-cell based immunity, for various types of cancers, with similar potency as complete genetic knockouts of these two PTPs, including some more nuanced combinatorial effects.

The crystallography is well done and appropriately reported, and does indeed help explain the striking effects of this compound. I am not a cancer biologist, but the experiments in cells and mice seem appropriate for the proposed hypotheses and include relevant controls, and the effects are quite dramatic. The amount of work on display here is impressive -- indeed, almost overwhelming (12 main figures, each with many panels & subpanels!). I do not have any major concerns with any key aspects of the paper.

Additional minor comments:

Line 202-204: I think you mean Fig. S5.

Fig. S5 caption: I think you mean Table 2.

For Table 2 (X-ray statistics): would be good to include MolProbity score, and include missing units for RMS bonds & angles, B-factors, etc.

In the future, please provide PDB model files, map files, and official validation reports along with the reviewer materials for crystal structures! In this case the information was sufficient to support the claims of the paper -- but this is a general recommendation for future work.

This paper should definitely be cited when discussing allosteric inhibitors of PTP1B:
<https://pubmed.ncbi.nlm.nih.gov/15258570/>

Can the authors comment on the relative potential for resistance mutations for this class of orthosteric inhibitors vs. for allosteric inhibitors?

Overall, this paper is a solid contribution to the fields of PTP inhibition and cancer drug discovery. It seems worthwhile for the community to have this record of how a very similar molecule to the AbbVie compound contributes to battling tumors, while said AbbVie compound continues in clinical trials.

Point-by-Point Rebuttal

Reviewer #1 (Remarks to the Author):

Major concern:

1. The largest concern about this work centers around its novelty. The authors present this work as an investigation of a compound directly related to a clinical candidate already in Phase 1 trials through Calico Life Sciences and AbbVie. This clinical candidate (ABBV-CLS-484) has been disclosed and discussed publicly (see https://aacrjournals.org/cancerres/article/82/12_Supplement/ND06/703170/Abstract-ND06-ABBV-CLS-484-An-active-site-PTPN2-N1). It is hard to imagine that the industrial developers of these compounds do not have very similar data to that presented in this manuscript, and it appears that at least some of these data were presented at the American Association for Cancer Research Annual Meeting in April of 2022 (based upon the abstract). It does not appear that the data have been published yet, however, somewhat mitigating this concern.

To the best of knowledge, the data from Abbvie has not been published. The data presented at the AACR annual meeting and potentially other meetings and associated abstracts do not constitute publication. Irrespective, our studies take advantage of a similar but different inhibitor and include a more extensive use of genetic models to define mechanisms of action in the context of immunogenic versus cold tumors, which in itself, is fundamentally important for the field.

Minor concerns:

“1. The authors are encouraged to use a consistent nomenclature for the enzymes they are targeting (i.e. either PTP1B/TCPTP or PTPN1/PTPN2).”

We thank the reviewer for this suggestion, but we would prefer to use PTPN2 in this instance as it is best known by this nomenclature in the immunology field, whereas PTP1B is generally recognised by this name.

“2. Several editorial mistakes are noted – the manuscript could use a thorough review for these types of minor mistakes. See, for example, lines 70, 106, 190, 431, 1105, 1172, etc.”

We apologise for these mistakes. We have carefully reviewed the manuscript and corrected these and other errors throughout.

“3. Similarly, the figure captions should be checked carefully. Most appear to be accurate, but the caption for Figure 1 is mislabeled (subfigures A, B, C are not as indicated).”

We apologise for the overt error in Fig 1 and have carefully checked and corrected all Figs/Fig legends.

“4. The compounds containing bromine should have two isotopes of nearly equal abundance, it would be a good idea to include the masses of both of the most prevalent isotopes in the HRMS data (lines 305, 731, etc.).”

We have now included masses for both bromine isotopes in the HRMS data as requested.

“Compound 182 should be clearly labeled in Scheme 1 – it would be helpful to include the structure of ABBV-CLS-484 here as well, or at least to reference Figure S1 early on (line 145 or so).”

We have now clearly labelled Compound 182 in Scheme 1, included the structure of ABBV-CLS-484 in Scheme 1 and referred to Fig. S1 on line 145 in the final paragraph of the introduction as requested.

“6. In the discussion, it could be helpful to be explicit that “In this study we focused on Compound 182 since the structure of AbbVie’s clinical candidates ... had not been revealed when the work began...”

We have now explicitly stated, “we focused on Compound 182 since the structures of AbbVie’s clinical candidates ABBV-CLS-484 and ABBV-CLS-579 were not revealed when our work began and....”

“7. The enzyme kinetic data seem to have been collected in an unusual way. The substrate DiFMUP is commonly used in assays of PTP activity and inhibition because it is a sensitive fluorogenic assay that can give continuous readouts of activity. It is not clear why the authors chose to carry out the assays in a discontinuous manner.”

With regard to the use of DiFMUP as a PTP substrate, both continuous and end-point assays have been utilised¹⁻⁴. For the sake of throughput, we measured all of our kinetic constants with the end-point measurements under steady-state conditions. To ensure robustness, we have optimised the assay conditions that yield similar enzymatic kinetic constants from both methods.

References:

- (1) Seale, A. P *et al.*, Development of an Automated Protein-Tyrosine Phosphatase 1B Inhibition Assay and the Screening of Putative Insulin-Enhancing Vanadium(IV) and Zinc(II) Complexes. *Biotechnol Lett* **2005**, 27 (4), 221–225. <https://doi.org/10.1007/s10529-004-7855-8>.
- (2) LaRochelle, J. R *et al.*, Structural Reorganization of SHP2 by Oncogenic Mutations and Implications for Oncoprotein Resistance to Allosteric Inhibition. *Nat Commun* **2018**, 9 (1), 4508. <https://doi.org/10.1038/s41467-018-06823-9>.
- (3) LaRochelle, J. R *et al.*, Identification of an Allosteric Benzothiazolopyrimidone Inhibitor of the Oncogenic Protein Tyrosine Phosphatase SHP2. *Bioorganic & Medicinal Chemistry* **2017**, 25 (24), 6479–6485. <https://doi.org/10.1016/j.bmc.2017.10.025>.
- (4) Chen, Y.-N. P. *et al.*, Allosteric Inhibition of SHP2 Phosphatase Inhibits Cancers Driven by Receptor Tyrosine Kinases. *Nature* **2016**, 535 (7610), 148–152. <https://doi.org/10.1038/nature18621>.

“The exact concentrations of DiFMUP should be included in the experimental section, as well (see line 883).”

The DiFMUP concentration used for the PTP1B and PTPN2 IC₅₀ measurements was 10 μM. This has now been included in the results as requested.

“The IC₅₀ curves should be provided, both for Compound 182 and for ABBV-CLS-484 (see line 549).”

The IC50 curves are now included in **Fig. S13**.

“For ABBV-CLS-484, the authors should indicate how they obtained this sample.”

ABBV-CLS-484 became available during the course of our studies at exorbitant prices. We purchased small amounts of ABBV-CLS-484 for *in vitro* studies from MedChemExpress. This has now been detailed in “Methods”.

“In addition, the inhibition of Compound 182 under standard conditions as well as under the conditions of the jump dilution should be shown in Figure 1B.”

The data has now been included in revised **Fig. 1b**.

“8. The selectivity of Compound 182 for PTPN1/PTPN2 over the panel of PTPs shown in Table 1 is impressive. However, for a true clinical candidate, the authors would need to investigate the activity of Compound 182 against other common targets (kinase panels, GPCR panels, etc.).”

Whether Compound 182 is ultimately a clinical candidate remains to be seen. Such analyses would be appropriate for ABBV-CLS-484 and ABBV-CLS-579 that are currently in trials. We may expect that such data would become available when Abbvie’s preclinical data is published. Nonetheless, it is important to note that these compounds are phosphotyrosine mimetics and would not be expected to bind to kinases and GPCRs. Although specifically assessing the activity of Compound 182 against other common targets extends beyond the scope of this study, we have established in our study that Compound 182’s effects on tumor growth and TILs *in vivo* phenocopy those evident when PTPN2 is deleted in AT3 tumor cells or when PTPN2 or PTP1B are deleted in T cells and that Compound 182 does not elicit any additional effects on tumor growth or TILs beyond those achieved by the deletion of PTPN2 in T cells. Therefore, our findings are consistent with the effects of Compound 182 on anti-tumor immunity and tumor growth being ascribed to the inhibition of PTP1B/PTPN2 in tumor cells and/or T cells.

“9. It would be helpful to explain the rationale behind dosing Compound 182 every 3 days – is this driven by PK/PD data, toxicity, etc.?”

This was not driven by PK/PD or toxicity data but by the limited availability of Compound 182, with each synthesis taking several months and yielding no more than 30 mg at a time. Nonetheless, we initially dosed Compound 182 **every 2 days versus 3 days**. Both repressed tumour growth and increased TILs, but the effects were greater when Compound 182 was administered every two days (**Fig. 4a** v/s **Fig, S6a**). Hence for all subsequent experiments (and all those in the main figures) we dosed every two days. We have now made this clearer in the revised manuscript. Dosing every two days did not result in any overt toxicities. We acknowledge that if Compound 182 was to proceed to clinical trials, PK/PD data would be essential. However, we deem such analyses are unnecessary for our preclinical studies, especially since dosing every two days achieved robust tumor repression.

10. The data indicating that Compound 182 does not promote inflammation or autoimmunity are quite solid (Fig 12), although the timeframe is fairly short. It would be interesting to investigate whether longer exposure to the compound results in a different outcome.

We have administered Compound 182 at 10 mg/kg every two days for 23 days i.e. a total of 8 doses. We see no overt signs of systemic inflammation or autoimmunity. This contrast with the overt systemic inflammation or autoimmunity accompanying the sustained hematopoietic specific deletion of PTPN2. Our studies establish for the first time that a therapeutic window exists for the combined systemic targeting of PTP1B/PTPN2 with SMIs. It is possible that the longer-term administration of Compound 182 or the use of higher doses of Compound 182 may ultimately result in systemic inflammation. However, this is to be expected for any therapeutic aimed at recruiting the immune system to combat cancer, as is the case for example for antibodies targeting the PD-1 and CTLA4 checkpoints. Ultimately this would be best addressed in a clinical setting. We have discussed this and the potential for driving CRS and autoimmunity especially in autoimmune prone individuals in the revised manuscript.

Reviewer #2 (Remarks to the Author):

“The paper by Liang and colleagues describes the development of a dual PTP1b/PTPN2 inhibitor and characterises its effects on immune cells and as an anti-cancer reagent. Overall the paper does a very thorough job of characterising this compound and demonstrates that it; a) is highly selective for PTPN2/PTP1b over other related phosphatases, b) promotes T cell activation via effects on TCR and cytokine receptor signalling, c) enhances T cell responses to tumours in vivo, d) reduces tumour growth in several different mouse models via effects primarily dependent on T cells, e) sensitises otherwise resistant tumours to anti-PD1 therapy, f) does not induce systemic inflammation despite potent anti-tumour function.

Overall, this is a very thorough paper with very convincing data that adds to evidence previously published by the same group suggesting that PTPN2 and or PTP1b are intracellular checkpoints for effective T cell anti-tumour responses and are potential therapeutic targets. The major novelty lies in the use of dual inhibitors that target both phosphatases; much of the biological effects of the inhibitor are as expected given the known functions of the phosphatases in T cells and previous studies showing that targeting either phosphatase alone through genetic means enhances anti-tumour immunity. However, the authors have done an excellent and thorough job and I struggle to find any significant weaknesses in the current work.”

We thank the reviewer for their supportive comments.

Reviewer #3 (Remarks to the Author):

“This paper reports in vitro and in cell studies of a small-molecule inhibitor specific to PTP1B and TCPTP (PTPN2) over all other PTP enzymes. The molecule is very similar to one currently in clinical trials for the same protein targets. A crystal structure of the compound is determined, which goes some way toward explaining the high specificity of this compound toward these two PTPs. A multitude of additional experiments in human cell lines and mouse models are employed to show that the inhibitor combats tumor growth by enhancing T-cell based immunity, for various types of cancers, with similar potency as complete genetic knockouts of these two PTPs, including some more nuanced combinatorial effects.

The crystallography is well done and appropriately reported, and does indeed help explain the striking effects of this compound. I am not a cancer biologist, but the experiments in cells and mice seem appropriate for the proposed hypotheses and include relevant controls, and the effects are quite dramatic. The amount of work on display here is impressive -- indeed, almost overwhelming (12 main figures, each with many panels & subpanels!). I do not have any major concerns with any key aspects of the paper.”

We thank the reviewer for their supportive comments.

Additional minor comments:

“Line 202-204: I think you mean Fig. S5.”

We apologise for this error; this has now been corrected.

“Fig. S5 caption: I think you mean Table 2.”

We apologise for this error; this has now been corrected.

“For Table 2 (X-ray statistics): would be good to include MolProbity score, and include missing units for RMS bonds & angles, B-factors, etc.”

The MolProbity score and missing units for RMS bonds & angles, B-factors etc have now been included in the table.

“In the future, please provide PDB model files, map files, and official validation reports along with the reviewer materials for crystal structures! In this case the information was sufficient to support the claims of the paper -- but this is a general recommendation for future work.”

We apologise for the inadvertent omission of the validation reports. These have now been included with the resubmitted manuscript. The provision of final data files is not routine for the field, but will be provided at the request of the journal.

“This paper should definitely be cited when discussing allosteric inhibitors of PTP1B: <https://pubmed.ncbi.nlm.nih.gov/15258570/>”

We apologise for this omission; we have now cited this article.

“Can the authors comment on the relative potential for resistance mutations for this class of orthosteric inhibitors vs. for allosteric inhibitors?”

We thank the reviewer for highlighting this. As with any other inhibitor targeting the active site of an enzyme, tumour cell resistance is possible. However, it is important to note that the anti-tumor effects of compound 182 can be ascribed largely to the inhibition of PTP1B/PTPN2 in T cells, where resistance would not occur. This has now been discussed in the revised manuscript.

“Overall, this paper is a solid contribution to the fields of PTP inhibition and cancer drug discovery. It seems worthwhile for the community to have this record of how a very similar molecule to the AbbVie compound contributes to battling tumors, while said AbbVie compound continues in clinical trials.”

We thank the reviewer for acknowledging the rigor and timely nature of our studies.